# Spatial distribution and factors associated with co-occurrence of anemia and undernutrition among children aged 6–59 months in East Africa

Altaseb Beyene Kassaw[1]*, Anissa Mohammed[2], Amare Muche[2], Asressie Molla[2]

**1** Department of Biomedical Science, College of Medicine and Health Science, Wollo University, Dessie, Ethiopia, **2** Department of Epidemiology and Biostatistics, School of Public Health, College of Medicine and Health Sciences, Wollo University, Dessie, Ethiopia

* altasebbeyene7@gmail.com

## Abstract

### Background

Anemia and undernutrition are major public health concerns affecting child growth, development, and survival, particularly in low-resource settings like East Africa. These conditions often occur together and exacerbate the health and developmental challenges of young children. Despite their high prevalence, no research has explored their co-occurrence and the spatial variations across East African countries. Hence, this study aimed to analyze the spatial distribution and identify predictors of the co-existence of anemia and under-nutrition among children aged 6–59 months in East Africa, utilizing recent Demographic and Health Survey data. Understanding the burden and factors affecting their co-occurrence is crucial for mapping geographic hotspot areas and identifying location-specific factors, which guide policymakers in developing targeted interventions.

### Methods and materials

This study analyzed the latest Demographic and Health Survey data from East African countries, focusing on children aged 6–59 months. It included those with complete outcome and geospatial data while excluding those with missing or inaccurate survey coverage. The primary outcome was the co-occurrence of anemia and undernutrition, which was defined as the presence of anemia together with at least one form of undernutrition, including stunting, wasting, or underweight. The data was managed and analyzed using STATA version 17, ArcGIS version 10.7.1, SaTScan v10.1, and MGWR version 2.2 software. Hotspot analysis was performed to identify high or low prevalence, and ordinary kriging was utilized for interpolation. Furthermore, the Bernoulli-based model was used to identify the most likely clusters of co-occurrences of these conditions by SaTScan analysis. Finally, the geographical

**Data availability statement:** All relevant data are within the manuscript and its Supporting information files.

**Funding:** The author(s) received no specific funding for this work.

**Competing interests:** The authors have declared that no competing interests exist.

**Abbreviations:** AICc, Akaike Information Criterion; DHS, Demographic and Health Surveys; GWR, Geographically Weighted Regression; LICs, Low-Income Countries; MGWR, Multiscale Geographically Weighted Regression; OLS, Ordinary Least Squares; SSA, Sub-Saharan Africa; UNICEF, United Nations International Children's Emergency Fund; WHO, World Health Organization.

weighted regression with the multiscale geographical weighted regression extension analysis was fitted to identify the spatially varying determinants.

## Results

The overall prevalence of co-occurrence of anemia and undernutrition was 26.08% (95% CI: 25.70–26.46%). The spatial analysis revealed that the distribution was non-random (p-value<0.001), with significant hotspot areas identified in northeast Ethiopia, northern and northwest Uganda, most parts of Burundi, northeast Tanzania, northern Zambia, central to southern Malawi, eastern Mozambique, and southern Madagascar. The primary clusters were concentrated in Burundi and southern Rwanda (RR = 3.48, LLR = 82.53; p-value <0.001). Spatial regression analysis identified several key predictors with geographic variation, including young maternal age (15–24 years), maternal anemia, no maternal education, poor household wealth, lack of health insurance, maternal tobacco use, multiple births, recent child diarrhea and fever, lack of Vitamin A supplementation, and smaller perceived birth size.

## Conclusion

The study highlights significant geographic disparities in the co-occurrence of anemia and undernutrition, with multiple maternal, child, and household factors contributing to the burden. These findings emphasize the need for region-specific public health interventions and future longitudinal research should explore additional confounders and refine regionally tailored approaches.

## Introduction

Anemia and undernutrition are both forms of malnutrition, which broadly refers to deficiencies, excesses, or imbalances in the intake of energy and nutrients [1]. Malnutrition is typically classified into three main categories: undernutrition, micronutrient-related malnutrition (deficiencies or excesses of essential vitamins and minerals such as iron), and conditions related to excess weight [1,2]. Each of the types of malnutrition is fundamentally different from the others [3], each is characterized by unique dietary causes and specific health consequences [3,4]. Undernutrition, which includes underweight, stunting (chronic malnutrition), and wasting (acute malnutrition), along with micronutrient deficiencies, is particularly prevalent among children under five [1,5]. Micronutrient deficiency, on the other hand, refers to inadequate intake of essential vitamins and minerals, including iron, vitamin B12, folate, and vitamin A [1], which can lead to health complications in children, including anemia [6].

The co-occurrence of undernutrition and anemia in children under five poses a serious public health challenge, particularly in low-income regions like sub-Saharan Africa (SSA), where poverty, food insecurity, and limited healthcare access prevail [7,8]. This combined burden has more severe consequences than either condition alone, hindering physical growth, cognitive development, and immune function,

and increasing vulnerability to infections, morbidity, and mortality [9,10]. Anemia, often caused by iron deficiency, worsens undernutrition's effects, creating a vicious cycle that undermines children's well-being and long-term health [10]. The underlying causes are multifaceted, involving nutritional deficits (especially iron, folate, and vitamin B12) [11,12], infections (such as malaria and parasitic diseases) [13,14] and socioeconomic factors [12]. Infections not only increase nutritional demands but also impair nutrient absorption and reduce iron bioavailability, further intensifying both conditions [15,16]. The high nutrient needs of early childhood, if unmet, lead to the body prioritizing energy for growth over red blood cell production, contributing to anemia [17]. [19,20]. Ultimately, undernutrition weakens immunity and heightens infection risk, depleting iron stores and worsening anemia, while anemia impairs development, reinforcing cycles of poor health and nutrition [18–20].

According to a WHO report, anemia affects approximately 40% of children globally, with the highest prevalence in regions such as SSA and South Asia [21]. In East Africa, approximately 75% of children have anemia, with the prevalence ranging between 35% and 80% [22]. The prevalence of undernutrition, including stunting, wasting, and underweight, also remains high in these areas. In 2022, it was estimated that globally, 22.3% of under-five children were stunted, and 6.8% were wasted [8]. Chronic undernutrition was estimated to affect 33.3% of under-five children in East Africa [23].

East African countries exhibit a high burden of childhood anemia and undernutrition [24]. Efforts such as improving dietary diversity, micronutrient supplementation, and community-based nutrition education have been made to alleviate this dual burden. In addition, interventions to improve nutritional intake and enhance healthcare access have been advocated [8,21,24,25]. However, progress has been slow, likely due to a limited context-specific understanding of the local factors driving the co-occurrence of anemia and undernutrition, as well as a lack of comprehensive policy frameworks that address these underlying determinants [8,24]. Moreover, the prevalence and distribution of anemia and undernutrition among under-five children in East African regions vary, partly due to regionally distinct factors, including socioeconomic disparities, environmental factors, regional food insecurity, and healthcare access [22,23,26,27]. Their interrelated risk factors, such as micronutrient deficiencies, cultural practices, inadequate breastfeeding practices, and limited access to clean water and sanitation, could create a synergistic effect for their dual burden [23,27,28]. In addition, seasonal and environmental factors, including agricultural yield variability and seasonal infections, further worsen these conditions, as seen in studies from Ethiopia [29,30] and Rwanda [10,31]. These issues are further exacerbated by geographic and demographic disparities, such as rural areas with limited healthcare access and higher rates of maternal illiteracy [32,33].

Despite extensive research on under-five malnutrition and anemia in East African countries, most studies have traditionally focused on either anemia or anthropometric failures in isolation, neglecting their combined occurrence and the regional disparities that influence both conditions. Additionally, many studies lack a spatial approach that captures region-specific factors using spatial analysis, limiting the understanding of region-specific predictors driving these conditions. Hence, this study aimed to assess the spatial distribution and associated factors of co-existing anemia and undernutrition among children aged 6–59 months in the East Africa Region. This would help map geographic hotspot areas and location-specific factors that guide policymakers in developing targeted interventions and allocating resources.

## Methods and materials

### Data source and study setting

This study used cross-sectional survey data from the Demographic and Health Surveys (DHS) conducted in East African countries between 2015 and 2023. Country classification was based on the United Nations geoscheme for Africa, which organizes countries into regional groupings for statistical and analytical purposes [34]. Under this classification, East Africa comprises 19 countries, including Burundi, Comoros, Djibouti, Ethiopia, Eritrea, Kenya, Madagascar, Malawi, Mauritius, Mozambique, Réunion, Rwanda, Seychelles, Somalia, Somaliland, Tanzania, Uganda, Zambia, and Zimbabwe. Countries

that are not traditionally classified as part of East Africa in some geographical frameworks, such as Zimbabwe, were included in accordance with this standardized classification to ensure analytical consistency.

Of the 19 countries classified as East African, DHS data were available for 13, while 6 (Djibouti, Somalia, Somaliland, Seychelles, Mauritius, and Réunion) do not. Among the 13 countries with DHS data, surveys from Eritrea (2002) and Comoros (2012) were conducted before 2015, and Kenya did not report anemia data. Consequently, the final analysis included DHS data from 10 countries with surveys conducted from 2015 onward (Table 1). The data was accessed from the DHS program website, https://dhsprogram.com/data/dataset, after permission had been obtained.

## Sampling procedure and sample size

The DHS surveys use a stratified, two-stage cluster sampling design to produce nationally representative estimates. Stratification is based on administrative regions and urban/rural residences to ensure diverse geographical and socioeconomic representations. In the first stage, primary sampling units (PSUs), often corresponding to census enumeration areas, are selected within each stratum using a probability proportional to size (PPS) approach. In the second stage, a fixed number of households (typically 20–30) are systematically selected from each cluster, and eligible individuals, including children aged 6–59 months, are surveyed. In this study, sampling weights were applied during the analysis to adjust for differences in selection probability and ensure representativeness across strata. The sample size included for the final analysis of this study is illustrated in Fig 1.

In the DHS survey, not all children in the surveyed population undergo biomarker testing. Therefore, only children with available hemoglobin measurements and complete anthropometric data were included in the analysis. Information about the survey's methodology is available online [35].

## Study population and data extractions

The study population comprised children aged 6–59 months preceding 5 years of the survey period across 10 East African countries, while the study population was children aged 6–59 months from the selected Enumeration Areas of the survey clusters. These children were those who lived with the respondent women aged 15–49 years at the time of the survey. Data was extracted from the DHS Kids Record dataset (KR file), which contains detailed information on children's health, nutrition, and demographic characteristics.

**Table 1. Study participants by country and corresponding DHS survey years for co-occurrence of anemia-undernutrition among children aged 6-59 months in East Africa.**

| S. N | Country | Survey year | Weighted frequency (%) |
|---|---|---|---|
| 1. | Burundi | 2016/17 | 5,582 (11%) |
| 2. | Ethiopia | 2016 | 8,4445 (16.64%) |
| 3. | Madagascar | 2021 | 5,0212 (9.89%) |
| 4. | Malawi | 2015/16 | 4,660 (9.18%) |
| 5. | Mozambique | 2022/23 | 3,382 (6.66%) |
| 6. | Rwanda | 2019/20 | 3,5245(6.94%) |
| 7. | Tanzania | 2022 | 4,322(8.52%) |
| 8. | Uganda | 2016 | 3,883 (7.65%) |
| 9. | Zambia | 2018 | 7,611 (14.99%) |
| 10. | Zimbabwe | 2015 | 4,328 (8.53%) |
| **Total** | 10 countries | 2015-2022 | 50,760 (100%) |

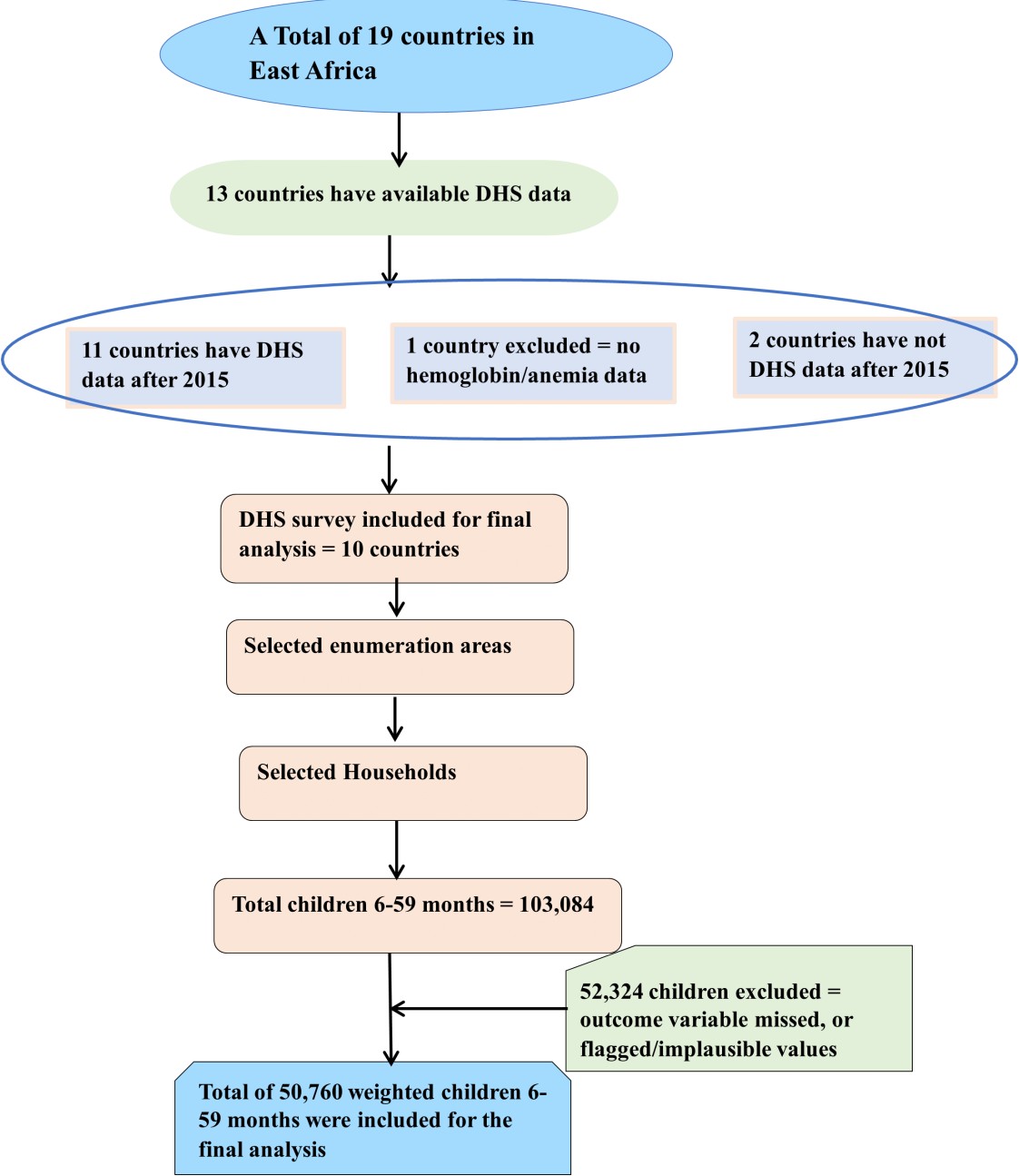

**Fig 1. The Flowchart illustrates the DHS data sampling process and final sample size selection.**

## Eligibility criteria

**Inclusion criteria.** The study included children aged 6–59 months who had complete data on hemoglobin levels and anthropometric measurements, specifically height-for-age, weight-for-height, and weight-for-age, for each participating country. Only children from countries with the most recent DHS datasets were considered for analysis

**Exclusion criteria.**  Children were excluded if they had missing or incomplete geospatial data or if their anthropometric measurements contained flagged, extreme, or implausible values, as defined by the WHO recommendation.

## Variable selections

All variables included in this study were selected based on peer-reviewed literature and the DHS datasets, ensuring that they are relevant to the research objectives. These variables were categorized and coded according to the guidelines in the DHS Recode Manual [36], which provides standardized coding procedures for all survey variables. This ensured consistency and comparability across countries and years of data collection. The variables were extracted from each country's most recent DHS datasets. The categorization included the following variables:

**Outcome variable.**  **Co-occurrence of anemia and undernutrition indices:** Defined as the presence of -

• Anemia (WHO criteria, hemoglobin level <11 g/dL) [1] **AND** one or more of the following undernutrition indices.

• Stunting (height-for-age Z-score < −2 SD) **OR,** Wasting (weight-for-height Z-score < −2 SD) **OR** Underweight (weight-for-age Z-score < −2 SD) [1].

**Independent variables.**  These variables were extracted from each country's most recent DHS datasets.

**Child-related factors included**: - Child's sex, age, child twin status, birth order, history of recent illness (acute respiratory infections, cough, fever, and diarrhea in the past 2 weeks), vitamin A supplementation in the past 6 months, iron-containing supplement, and deworming in the past 6 months, preceding birth interval, place of delivery, perceived size of child at birth.

**Parental/caregiver-related factors include**:– educational status, maternal literacy, maternal age, marital status, maternal employment, maternal anemia status, maternal BMI, and maternal smoking/ tobacco use.

**Household characteristics:** Household size, household wealth status, availability of health insurance, number of under-five, sex of the household head, household media exposure, and types of cooking fuel.

**Environmental and community level factors**: Residence, source of drinking water quality**,** type of toilet facility/sanitation facility, problems in accessing health care, and altitude.

## Operational definitions

**Media exposure**: It was calculated by combining the four variables, such as watching television, listening to the radio, reading newspapers, and using the internet. Then categorized as having media exposure if a mother has been exposed to at least one of the four, and not if she had no exposure to any of the media sources [37].

**Iron-containing supplement**: – was categorized based on the DHS guiding manual as "Yes" or "No". if given iron-containing supplements in the 12 months preceding the interview (who were given iron tablets or syrups or multiple micronutrient powders), coded as "Yes, otherwise "No" [36].

**Maternal anemia: –** Maternal anemia was defined according to WHO guidelines, with hemoglobin <12 g/dL for non-pregnant women and <11 g/dL for pregnant women. Hemoglobin values were adjusted for altitude and smoking, where applicable.

**Problems with accessing health care:** – was coded based on the DHS guiding manual. As "Not big problem" and "Big problem". If the respondents had serious problems in accessing health care when their child is sick, by type of problem (getting permission to go for treatment, getting money for treatment, distance to the health facility-big problem, or not wanting to go with the child alone), categorized as Big problem", otherwise "Not big problem" [36].

**Source of drinking water quality: –** was categorized based on the DHS guiding manual as improved, unimproved, and other sources. Improved drinking water sources: piped water, public taps/standpipes, boreholes or tubewells, protected dug wells, protected springs, rainwater, and bottled water (if an improved water source was used for cooking and handwashing) [36].

**Type of toilet facility/sanitation facility quality**: Was dichotomized as improved or unimproved. Non-shared facilities that flush/pour flush to a piped sewer system, septic tank, or pit latrine, ventilated improved pit latrine, pit latrine with a slab, and composting toilet were categorized as having improved sanitation. Facilities that flushed to a known location but not to a sewer system, septic tank, or pit latrine (i.e., those that flush to "somewhere else"), pit latrine – without slab/open pit, bucket toilet, hanging toilet/latrine, and others were classified as unimproved [36].

## Data quality control

The DHS program ensures high data quality through rigorous quality assurance procedures, starting from survey design to data processing and validation. Furthermore, several control measures were implemented throughout this research before any analysis. Datasets were checked for missing values, particularly for key variables like anemia and undernutrition indicators, with appropriate methods for imputation or exclusion based on the pattern of missingness. Outlier detection techniques were used to identify and correct inconsistent or implausible values. Data from different East African countries were cross-checked for consistency in variable definitions and standardized for meaningful comparisons. Weighting adjustments were applied to account for the complex sampling design, ensuring representativeness.

## Data management and data analysis

The dataset for each country was extracted and cleaned to remove missing and inconsistent observations. The cleaned datasets from 10 countries were appended into a single dataset for further analysis after harmonizing variable names and coding. STATA version 17 was utilized for data extraction, recording, and analysis. The data was thoroughly checked for completeness and appropriately weighted before statistical analysis. Sampling weights were applied to account for potential differences in response rates and to ensure that the survey results were representative at both the national and regional levels. These weights were also adjusted for the non-proportional allocation of the sample across different regions, thereby enhancing the accuracy and generalizability of the findings.

Descriptive statistics, including percentages and frequencies, were applied to summarize the background characteristics of the study participants. ArcGIS version 10.7.1, SaTScan v10.1, and MGWR version 2.2 were utilized for spatial data analysis.

**Spatial analysis.** *Spatial autocorrelation analysis*: Global Moran's I measure was used to verify whether the co-occurrence of anemia and undernutrition was clustered, dispersed, or randomly distributed in East Africa. Global Moran's I calculates the Moran's I Index value, Z score, and p-value. Moran's I index close to −1 indicates that cases of co-occurrence of anemia and undernutrition were dispersed, whereas an I index close to 1 indicates that cases were clustered, and zero indicates that cases are distributed randomly [38]. A statistically significant Z-score and p-value ≤ 0.05 indicate the existence of clustering. A statistically nonsignificant Moran's I value (p value> 0.05) indicates that cases of co-occurrence of anemia and undernutrition were randomly distributed throughout the region.

*Hotspot analysis*: Hotspot analysis was performed by calculating the GI* statistic (Getis-OrdGi* statistic) for each area and statistical outputs with high GI* indicate "hotspot" areas, while low GI* indicates "cold spot" areas. The GI* statistic is a Z-score. A high Z-score and small p-value for a feature suggest a significant hotspot. Significant cold spot areas are indicated by a small p-value and a low negative Z-score. The intensity of the clustering increases with a greater (or lower) Z- score. A Z-score near zero means no spatial clustering. Thus, using this methodology, statistically significant regional clusters of high values (hotspots) and low values (cold spots) of children aged 6–59 months having co-occurrences of anemia and undernutrition were identified.

*Spatial scan statistical analysis*: Bernoulli-based spatial scan analysis was conducted to identify the geographic locations of statistically significant clusters for the co-occurrence of anemia and undernutrition among children aged 6–59 months, utilizing SaTScan™ software version 10.1.3. The spatial scan statistic employs a circular scanning window that

encompasses statistically significant spatial clusters related to the co-occurrence of anemia and undernutrition. Children aged 6–59 months who exhibited co-occurrences of anemia and undernutrition were included in the model as cases, while children without these co-occurrences were placed in the control group. A binary variable with values of 0 and 1 was utilized. The circle with the highest likelihood represented the primary cluster (most likely cluster). The other significant clusters were classified as secondary clusters, which were generated by automatically removing the primary clusters in the SaTScan software, ensuring no geographic overlap during the analysis. These clusters were then ranked according to their likelihood ratio test statistics [39]. The relative risk (RR), location radius, and log-likelihood ratio (LLR) test statistics with p-values were presented for each cluster.

*Spatial interpolation*: The Ordinary Kriging spatial interpolation technique was used for predicting the co-occurrence of anemia and undernutrition among children aged 6–59 months in unobserved areas, based on known values of the surrounding sampled points.

**Spatial regression.** *Ordinary least squares (OLS) regression*: Initially, exploratory regression analysis was performed to identify candidate explanatory variables for inclusion in the model. The model systematically tested multiple combinations of predictor variables to evaluate their significance and explanatory power. Variables that showed statistical significance ($p < 0.05$), improved model fitness, and adhered to statistical assumptions, such as non-multicollinearity, were chosen for further analysis. After variable selection, ordinary least squares (OLS) were employed to examine the global relationship between the co-occurrence of anemia and undernutrition and the selected predictors. It assumes a stationary and constant relationship across space, meaning that the relationships do not vary spatially (i.e., it predicts only one coefficient per independent variable throughout the research area). Subsequently, model performance was evaluated using the adjusted R-squared and the corrected Akaike Information Criterion (AICc), along with assessments of model assumptions, including the Variance Inflation Factor (VIF), the autocorrelation of residuals, the anticipated sign for coefficients, and Koenker and Jarque-Bera test statistics.

*Geographically weighted regression*: To account for spatial heterogeneity, Geographically Weighted Regression (GWR) with Multiscale Geographically Weighted Regression (MGWR) extension was implemented using the same sets of explanatory variables. Unlike OLS, which assumes a global relationship between independent and dependent variables, GWR allows coefficients to vary across geographic locations, providing localized parameter estimates. It assumes that the explanatory factors and the outcome have nonstationary spatial correlations, with coefficients changing at about the same pace throughout the research region. Moreover, it only employs a single constant bandwidth.

The MGWR model was applied to calibrate the parameter estimates of GWR using MGWR version 2.2. The MGWR is an extension of GWR that allows different processes to vary over space at various spatial scales. Rather than using a single constant bandwidth across the entire study region, this model employs multiple bandwidths, allowing the relationship between the dependent and independent variables to vary spatially across various spatial scales. The adaptive bi-square kernels were utilized for geographical weighting to estimate local parameter estimates. The golden section search method was applied to determine the optimal bandwidth size based on the AICc.

The results of the OLS, GWR, and MGWR models were compared to assess the improvement in model fit and the spatial variation in predictor relationships. Model performance was evaluated using metrics such as AICc and Adjusted R-squared. The spatial distribution of the local coefficients from MGWR was mapped to visualize the spatial variation in the relationships, as the model provided a better fit.

### Ethics statement

This study is a secondary data analysis based on publicly available, de-identified data from the MEASURE DHS Program. The DHS surveys are implemented by ICF International and partner organizations with support from USAID and receive ethical approval from the Institutional Review Board (IRB) of ICF International, as well as national ethics committees in

each participating country. Informed consent was obtained from all participants by the DHS Program before data collection. The datasets are fully anonymized and contain no identifiable personal information.

The authors accessed the data on October 29, 2024, following approval from the DHS Program to use data from East African countries. As this study involved no direct contact with participants and used only anonymized, publicly available data, no additional ethical approval or consent was required. Permission to use the dataset was granted by the DHS Program (https://dhsprogram.com/).

## Results

### Background characteristics of the study participants

The study included a total of 50,760 weighted children aged 6–59 months from ten East African countries, with the highest proportion from Ethiopia and the lowest from Mozambique. Among the participants, 21,666 (42.7%) were aged 36–59 months. The sex distribution was nearly balanced, with 25457(50.2%) males. Additionally, 42,577 (83.9%) of the participants had not experienced diarrhea, and 39,902 (78.6%) had not reported fever in the two weeks before the survey. About 46.2% and 60.2% of the children received deworming and vitamin A in the last 6 months preceding the survey, respectively. More than half of the children, 28044 (55.2%), were from households having media exposure. Among the total participants, 32087(63.2%) and 28438 (56.0%) were from households having big problems in accessing health care and unimproved sanitary facilities, respectively.

Nearly half of the children's mothers (48.0%) were aged 25–34 years, and 47.4% had completed primary education. Most of the mothers, 43761 (86.2%), were married/living with a partner, and the majority, 29432(58.0%) fell under the currently working category. In addition, 29432(58.0%) mothers were currently employed (Table 2).

### Prevalence of co-occurrence of anemia and undernutrition

The overall prevalence of anemia among children aged 6–59 months in East Africa was estimated at 26.08% (95% CI: 25.70–26.46%), with variations among countries. The highest prevalence was recorded in Burundi at 40.63%, followed by Mozambique at 33.80%, while the lowest was observed in Zimbabwe at 13.36% (Fig 2).

Further breakdown by age and sex showed that children aged 12–23 months and males had higher prevalence rates (32.19% and 28.57%, respectively) compared to their counterparts.

From the total weighted sample, 43.12% (95% CI: 42.69–43.55%) of children were undernourished, with the prevalence of undernutrition indicators being 38.84% for stunting, 17.40% for underweight, and 4.90% for wasting. Depending on anemia status, 40.28% of children were anemic, with 25.51%, 27.85%, and 2.05% classified as having mild, moderate, and severe anemia, respectively. The co-occurrence of anemia with each undernutrition indices is depicted in Fig 3.

### Spatial distribution of co-occurrence of anemia and undernutrition

The spatial distribution of co-occurrence of anemia and undernutrition among children 6–59 months of age was nonrandom in East Africa (Fig A in S1 File). Across the region, it was spatially clustered with a Global Moran's value of 0.219194 (p<0.001) and a Z-score of 94.836630, which indicated that there is less than 1% probability of the cluster due to the result of chance (Fig 4).

Incremental spatial autocorrelation was performed to identify the maximum clustering. At a starting distance of 207644.62 meters, a total of 20 distance bands were detected, and the maximum clustering was noted at 414419.50 meters (z-value=66.09). Incremental autocorrelation results are presented in Fig B in S1 File.

**Hot spot analysis of the co-occurrence of anemia and undernutrition.** The hot spot analysis was performed to identify high-risk areas for the co-occurrence of anemia and undernutrition. The red color, which indicates the significant hotspot areas (high prevalence of co-occurring anemia and undernutrition), was found in Northeast Ethiopia, Northern and Northwest Uganda, most parts of Burundi, Northeast Tanzania, Northern Zambia, Central to Southern Malawi, much

**Table 2. Background characteristics of the study participants.**

| Variables | | Weighted frequency (N) | Percentage (%) |
|---|---|---|---|
| Child age (in months) | 6-11 | 6109 | 12.0 |
| | 12-23 | 11821 | 23.3 |
| | 24-35 | 11165 | 22.0 |
| | 36-59 | 21666 | 42.7 |
| Child sex | Male | 25457 | 50.2 |
| | Female | 25303 | 49.8 |
| Residence | Urban | 10208 | 20.1 |
| | Rural | 40553 | 79.9 |
| Maternal age | 15-24 | 13875 | 27.3 |
| | 25-34 | 24343 | 48.0 |
| | 35-49 | 12541 | 24.7 |
| Highest maternal educational level | No education | 13732 | 27.1 |
| | Primary | 24039 | 47.4 |
| | Secondary | 11596 | 22.8 |
| | Higher | 1393 | 2.7 |
| Maternal literacy | Cannot read at all | 20767 | 40.99 |
| | Able to read parts of a sentence | 5474 | 10.80 |
| | Able to read the whole sentence | 24428 | 48.21 |
| Currently employed | No | 21328 | 42.0 |
| | Yes | 29432 | 58.0 |
| Current Marital status | Never in union | 2384 | 4.7 |
| | Married/living with partner | 43761 | 86.2 |
| | Divorced/widowed/separated | 4615 | 9.1 |
| Maternal BMI | Underweight | 4893 | 9.6 |
| | Normal | 30167 | 59.4 |
| | Overweight/Obesity | 7951 | 15.7 |
| | Missing | 7749 | 15.27 |
| Maternal anemia | Anemic | 15593 | 30.89 |
| | Not anemic | 34888 | 69.11 |
| Husband/partner's education level | No education | 17299 | 34.1 |
| | Primary | 20032 | 39.5 |
| | Secondary | 11290 | 22.2 |
| | Higher | 2139 | 4.2 |
| Drinking water source quality | Unimproved | 16834 | 33.16 |
| | Improved | 33926 | 66.84 |
| Sanitation facility | Unimproved | 29118 | 57.36 |
| | Improved | 21642 | 42.64 |
| Household media exposure | No Media Exposure | 22716 | 44.8 |
| | Have Media Exposure | 28044 | 55.2 |
| Family size | 2-5 | 24780 | 48.8 |
| | >5 | 25980 | 51.2 |
| Number of children 5 years and under in the household | 1 | 20252 | 39.9 |
| | 2-3 | 29026 | 57.2 |
| | >=4 | 1482 | 2.9 |
| Sex of household head | Male | 39747 | 78.3 |
| | Female | 11013 | 21.7 |

*(Continued)*

**Table 2.** (Continued)

| Variables | | Weighted frequency (N) | Percentage (%) |
|---|---|---|---|
| Types of cooking fuel | Clean fuel/Not cooked | 2233 | 4.4 |
| | Solid fuel | 47837 | 94.2 |
| | Other | 691 | 1.4 |
| Tobacco use/ smoking cigarettes mother | No | 49261 | 97.0 |
| | Yes | 1500 | 3.0 |
| Problems in accessing health care | Not big problem | 18673 | 36.8 |
| | Big problem | 32087 | 63.2 |
| Health insurance | No | 45362 | 89.4 |
| | Yes | 5398 | 10.6 |
| Child twin status | Single | 49461 | 97.4 |
| | Multiple | 1299 | 2.6 |
| Birth order number | Firstborn | 11134 | 21.93 |
| | 2-3rd born | 18689 | 36.82 |
| | 4-5th born | 11375 | 22.41 |
| | >=6th born | 9563 | 18.84 |
| Preceding birth interval (in months) | First birth | 11231 | 22.1 |
| | <24 | 6585 | 13.0 |
| | 24-35 | 12161 | 24.0 |
| | >35 | 20784 | 40.9 |
| Place of delivery | Home delivery | 19679 | 34.8 |
| | Institutional delivery | 33081 | 65.2 |
| Perceived size of child at birth | Smaller than average/very small | 7961 | 15.7 |
| | Average | 24097 | 47.5 |
| | Larger than average/ very large | 14590 | 28.7 |
| | Missing | 4113 | 8.1 |
| Delivery by caesarean section | No | 48257 | 95.1 |
| | Yes | 2503 | 4.9 |
| Breastfeeding status | Ever breastfeed, | 2,820 | 64.66 |
| | Never breastfeed | 1,015 | 2.00 |
| | Currently breastfeed | 16,925 | 33.34 |
| Had diarrhea in the last 2 weeks | No | 42577 | 83.9 |
| | Yes, | 8183 | 16.1 |
| Had a fever in the last 2 weeks | No | 39902 | 78.6 |
| | Yes | 10858 | 21.4 |
| Had ARI or/and cough | No | 36652 | 72.2 |
| | Yes | 14108 | 27.8 |
| Iron supplementation status | No | 45098 | 88.8 |
| | Yes | 5662 | 11.2 |
| Vitamin A in the last 6 months | No | 20212 | 39.8 |
| | Yes | 30548 | 60.2 |
| Deworming in the last 6 months | No | 27312 | 53.8 |
| | Yes | 23448 | 46.2 |
| Cluster altitude in meters | <1000 | 14883 | 29.32 |
| | 1000-2000 | 31249 | 61.56 |
| | 2001-3500 | 4552 | 9.12 |

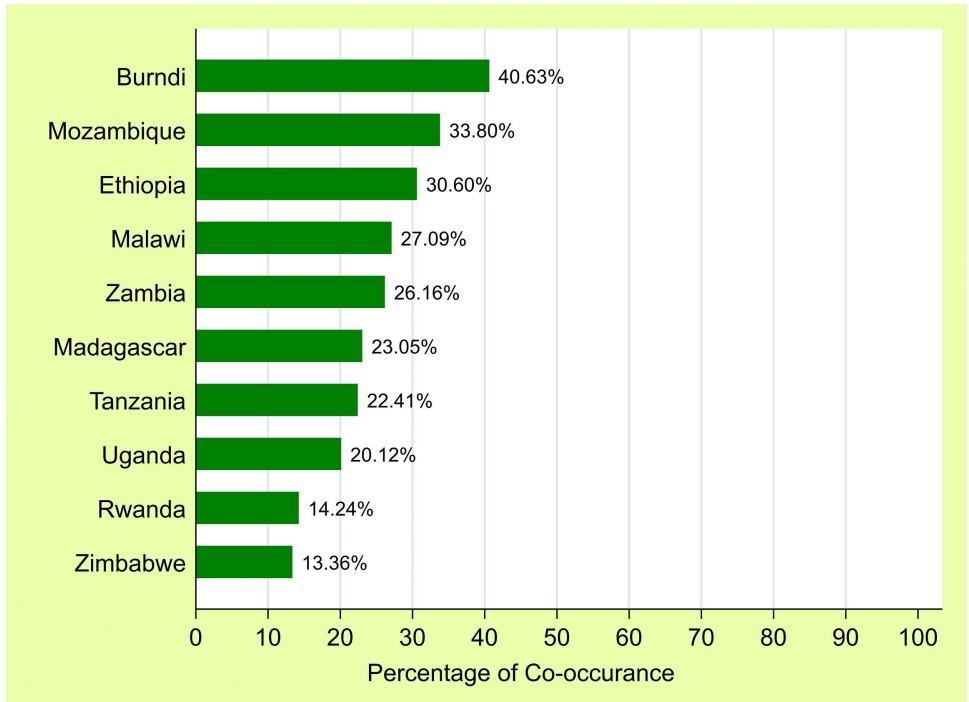

**Fig 2. Distribution of co-occurrence of Anemia and undernutrition in East Africa by country.**

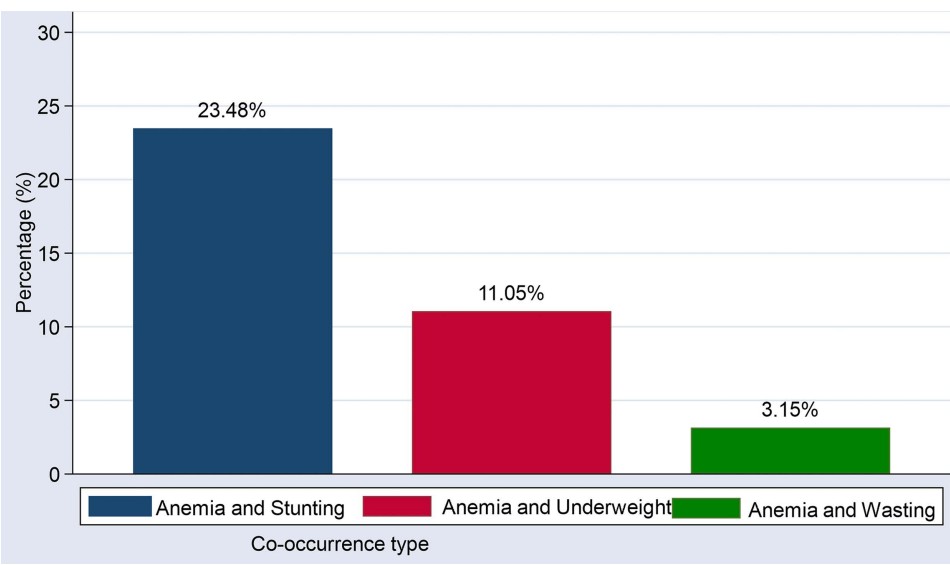

**Fig 3. Co-occurrence of undernutrition indices and anemia among children 6-59 months in East Africa.**

of Eastern Mozambique, and Southern Madagascar. Whereas the blue color indicates the cold spot area (low prevalence of co-occurrence of anemia and undernutrition), which was found in Central Ethiopia, Southeast and Central to Eastern Uganda, most parts of Zimbabwe, and the Southern region of Mozambique (Fig 5).

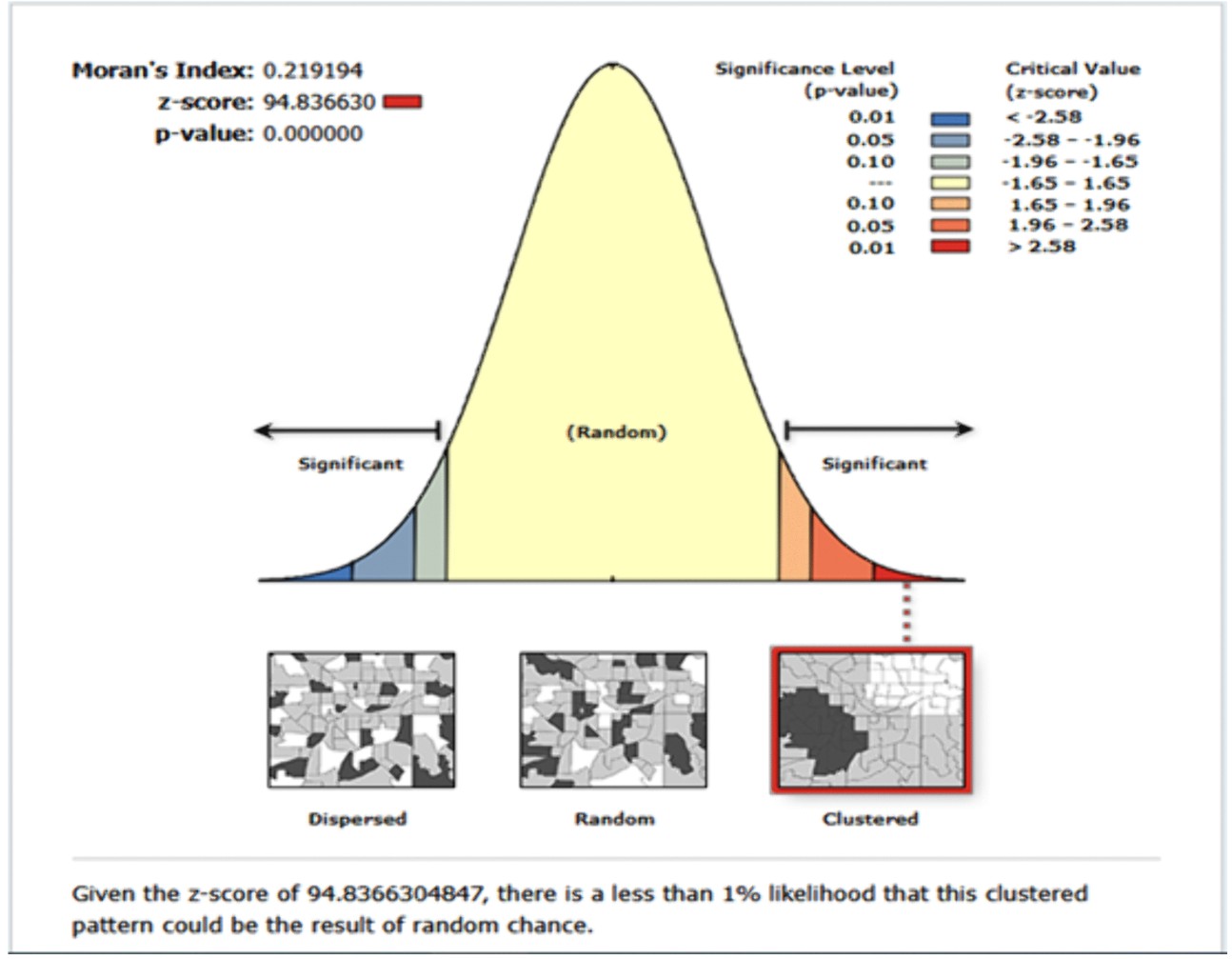

**Fig 4. Spatial autocorrelation analysis of co-occurrences of anemia and undernutrition among children aged 6-59 months in East Africa.**

**Spatial interpolation of the co-occurrence of anemia and undernutrition.** Ordinary Kriging interpolation in unsampled enumeration areas indicated a high predicted prevalence of co-occurring anemia and undernutrition in eastern and southern Ethiopia, northern Uganda, northeastern Burundi, north of Zambia, the southern to eastern regions of Mozambique, and northern Madagascar (Fig C in S1 File).

**Spatial Sat Scan analysis.** A total of 7 significant spatial windows were identified, with 146 locations falling within the most significant primary cluster. The spatial window for the primary clusters was located in most parts of Burundi and Southern Rwanda, centered at (3.22°S, 30.54°E) with a 97.42 km radius. Children in this cluster were 3.48 times more likely to have co-occurrence of anemia and undernutrition compared with those outside this window (RR = 3.48, LLR = 82.53; P-value<0.001).

The second most likely SaTScan window covered mainly the Northeast and Southeast parts of Mozambique, which centered at 15.25°S, 39.35°E/271.52 km/radius. Children within this spatial window were 3.37 times more likely to have co-occurrence of anemia and undernutrition, compared to those outside this window (RR = 3.37, LLR = 42.76; P-value<0.001) (Fig 6). Spatial Sat Scan results are summarized in Table A of the S1 File.

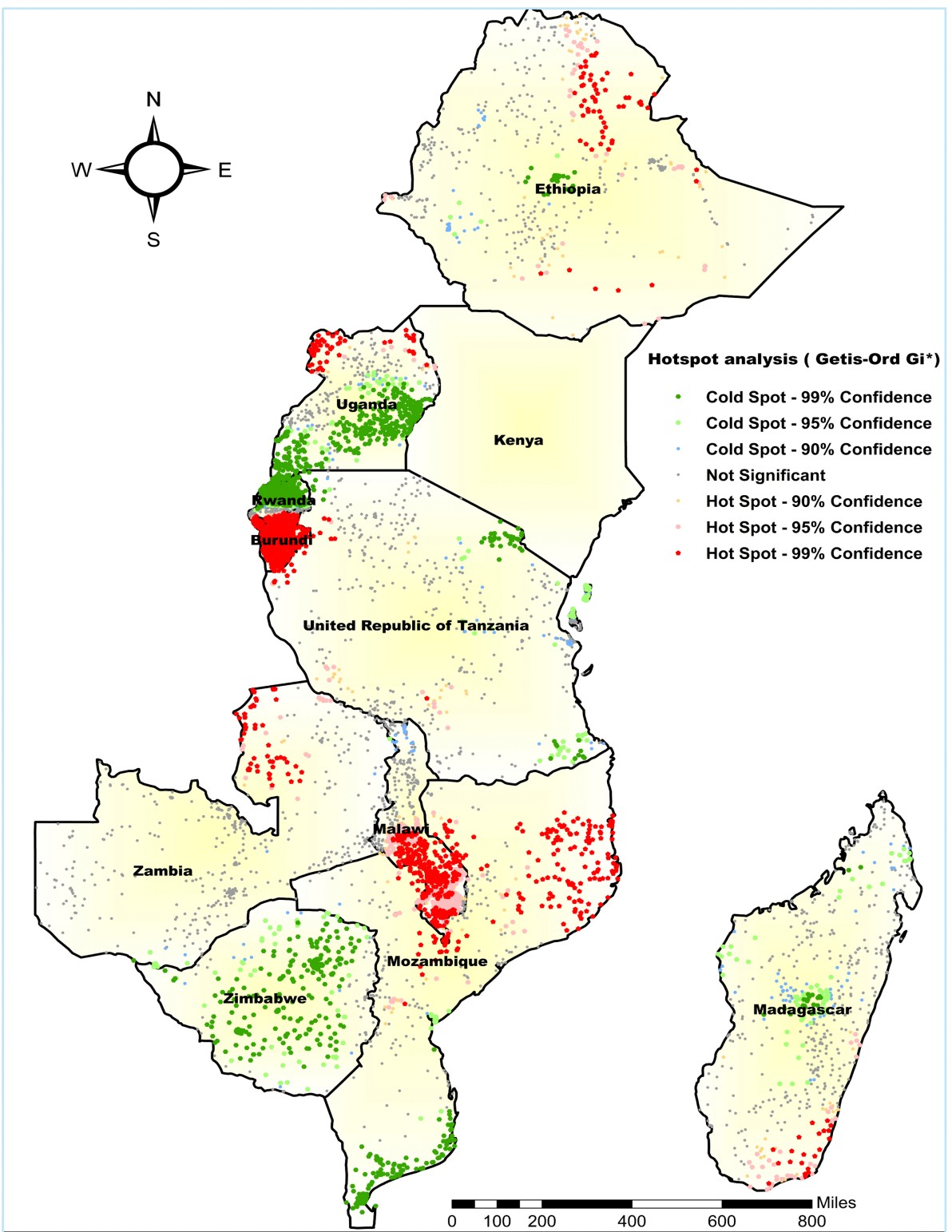

**Fig 5. Hotspot analysis of co-occurrence of anemia and undernutrition in East Africa among children 6-59 months of age in East Africa.**
**Source**: Administrative boundary shapefile obtained from OpenAfrica (https://open.africa/dataset/africa-shapefiles). [**Note**: The authors utilized the shapefile solely as a basemap, performing all spatial processing, analysis, visualization, and modifications for illustrative and analytical purposes only].

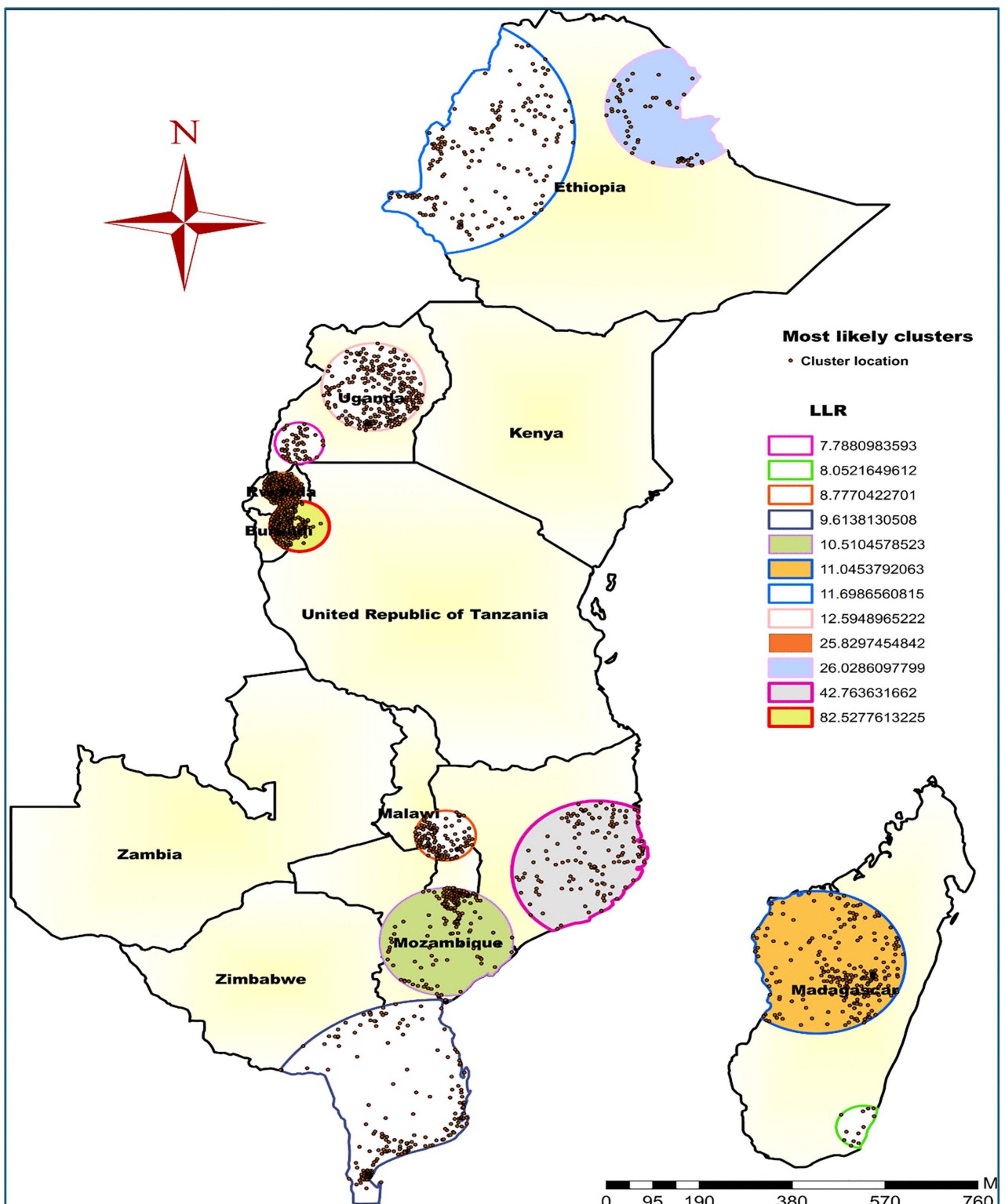

**Fig 6. Spatial SaTscan analysis of co-occurrence of anemia and undernutrition among children aged 6-59 months in East Africa. Source**: Administrative boundary shapefile obtained from OpenAfrica (https://open.africa/dataset/africa-shapefiles). [**Note**: The authors utilized the shapefile solely as a basemap, performing all spatial processing, analysis, visualization, and modifications for illustrative and analytical purposes only.

## Spatial regression analysis for factors associated with spatial variations of co-occurrence of anemia and undernutrition among children aged 6–59 months in East Africa

**Exploratory regression analysis.** On exploratory regression analysis, multiple combinations of predictors were tested, and the following variables were retained for subsequent spatial analysis. Proportion of- being from an anemic mother, absence of vitamin A supplementation in the past six months, recent diarrhea or fever, maternal tobacco use, maternal age 15–24 years, lack of maternal education, being from an overweight mother, belonging to the poorest or poorer households, lack of health insurance, no media exposure, being a female child, child age 12–23 months, multiple births, birth interval of less than 24 months, and smaller or very small perceived size at birth.

**Ordinary least squares regression analysis and diagnostic check.** The OLS results indicated a significant negative association of maternal overweight/obesity and female sex with the outcome, while all other predictors showed positive associations. No multicollinearity was detected, as all variance inflation factors (VIF) values were below 7.5. As the Koenker (BP) statistics were significant, robust probabilities were used to evaluate coefficient significance, and all coefficients remained statistically significant ($p < 0.05$). The model explained 53.89% of the variation in the co-occurrence of anemia and undernutrition (Adjusted $R^2 = 0.5389$; AICc $= 15688.91$). However, the spatial autocorrelation test (Moran's I $= 0.11$, $p < 0.01$) revealed significant spatial clustering of residuals. The significant Koenker (BP) statistic further indicated non-stationarity in relationships between predictors and outcomes, justifying the use of a spatially adaptive modeling approach to account for spatial dependence (Tables 3 and 4).

**Geographically weighted regression analysis (GWR) and the Multiscale extension (MGWR).** Results from the OLS regression indicated significant Koenker (BP) and Jarque-Bera statistics, suggesting non-stationarity among variable relationships and non-normal residual distribution, respectively. To address these issues, spatially nonstationary local models, GWR and MGWR, were applied using the same explanatory variables as in the OLS model. Both models outperformed the global OLS model in explaining the spatial variation of anemia and undernutrition co-occurrence among children aged 6–59 months in East Africa. The adjusted $R^2$ increased from 53.89% (OLS) to 55.7% (GWR) and 64.3%

**Table 3.** The OLS model parameter estimates for the co-occurrence of anemia and undernutrition among children 6-59 months in East Africa.

| Variable | Coefficient [a] | SE | t-Stat | Probability [b] | Robust_SE | Robust_t | Robust_Pr [b] | VIF [c] |
|---|---|---|---|---|---|---|---|---|
| Intercept | 0.0596 | 0.011 | 5.3264 | 0.00000* | 0.010047 | 5.937384 | 0.000000* | -------- |
| Maternal age 15–24 | 0.0282 | 0.0110 | 2.5436 | 0.010985* | 0.013631 | 2.069059 | 0.038571* | 1.1452 |
| No maternal education | 0.1452 | 0.0107 | 13.5289 | 0.00000* | 0.01200 | 12.092 | 0.0000* | 1.5044 |
| Maternal overweight | −0.0798 | 0.0107 | −7.4265 | 0.00000* | 0.01208 | −6.6034 | 0.0000* | 1.2763 |
| Maternal anemia | 0.1325 | 0.0095 | 13.8210 | 0.00000* | 0.01156 | 11.461 | 0.0000* | 1.1097 |
| Poorest household | 0.0455 | 0.0105 | 4.3012 | 0.00002* | 0.01127 | 4.0389 | 0.0000* | 1.4545 |
| Poorer household | 0.0596 | 0.0119 | 5.0015 | 0.00000* | 0.01341 | 4.4407 | 0.0000* | 1.1702 |
| No media exposure | 0.0343 | 0.0099 | 3.4559 | 0.000568* | 0.011290 | 3.040040 | 0.002388* | 1.7906 |
| Tobacco user/ smoker mother | 0.0821 | 0.0253 | 3.2396 | 0.00122* | 0.03339 | 2.4584 | 0.0139* | 1.0561 |
| No health insurance | 0.0295 | 0.0099 | 2.9713 | 0.002987* | 0.009219 | 3.204483 | 0.001376* | 1.2417 |
| Female sex | −0.0380 | 0.0110 | −3.4518 | 0.00057* | 0.013200 | −2.8815 | 0.0038* | 1.0148 |
| Child age 12–23 | 0.0694 | 0.0137 | 5.0634 | 0.00000* | 0.01704 | 4.0725 | 0.0000* | 1.0288 |
| Multiple birth | 0.1493 | 0.0322 | 4.6234 | 0.00000* | 0.03755 | 3.9759 | 0.0000* | 1.0081 |
| Birth interval< 24 months | 0.0450 | 0.01764 | 2.5503 | 0.010778* | 0.021678 | 2.076243 | 0.037901* | 1.0631 |
| Having Recent diarrhea | 0.0430 | 0.0150 | 2.8557 | 0.004314* | 0.018723 | 2.297322 | 0.021618* | 1.1718 |
| Having recent fever | 0.0949 | 0.0114 | 8.3214 | 0.000000* | 0.014389 | 6.597422 | 0.000000* | 1.2036 |
| Not having vit-A supplement | 0.0259 | 0.0091 | 2.84721 | 0.00443* | 0.010704 | 2.420895 | 0.015497* | 1.1850 |
| Smaller perceived birth size | 0.0480 | 0.0176 | 2.72995 | 0.00635* | 0.02297 | 2.0920 | 0.0364* | 1.0278 |

**Table 4. The OLS model diagnostics parameter for the co-occurrence of anemia and undernutrition among children 6-59 months in East Africa.**

| Diagnostic Criteria | Values | Pr- |
|---|---|---|
| Akaike's Information Criterion (AICc) | 15688.908 | – |
| Multiple R-Squared | 0.56073 | – |
| Adjusted R-Squared | 0.5389 | – |
| Joint F-Statistic | 98.896982 | 0.000000* |
| Joint Wald Statistic | 1625.778597 | 0.000000* |
| Koenker (BP) Statistic | 162.886743 | 0.000000* |
| Jarque-Bera Statistic | 1336.501895 | 0.000000* |

(MGWR), while the AICc decreased from 15,688.91 (OLS) to 15,274.92 (GWR) and 15098.065(MGWR), indicating improved model fit. Among the three, the MGWR model demonstrated the best performance and was identified as the most appropriate for further analysis (Table 5).

The result of the GWR analysis revealed spatial variations in the relationship between the co-occurrence of anemia and undernutrition prevalence and key predictor variables. Summary statistics of GWR coefficients are presented in Table A of S2 File.

The analysis from MGWR in this study revealed that the proportion being from overweight mothers, children aged 12–23 months, and not having health insurance had a broader bandwidth, suggesting their influence operated across larger regions, while having the remaining variables had a relatively narrower bandwidth, indicating a more localized effect (Table 6).

**Mapping the parameter estimates.** Given the superior performance of the MGWR model, parameter estimates from this model were mapped to visualize the spatial relationships between the co-occurrence of anemia with undernutrition and the selected predictors. Coefficient estimates were displayed using a color gradient from bright red (strong association) to green (weak association).

For the proportion of maternal anemia, the MGWR model revealed clear spatial variability in the beta (β) coefficients, showing a generally positive association (mean β = 0.151). The highest coefficient values (β = 0.2924–0.4143) were observed in Southwest and Northwest Ethiopia, Northern and Eastern Burundi, and Southeast and Southwest Madagascar, indicating that a one-unit increase in maternal anemia was associated with a 29.24–41.43 percentage point rise in the co-occurrence of anemia and undernutrition in these regions. Conversely, the lowest coefficient values

**Table 5. Model diagnostic information and the model's performance for the co-occurrence of anemia and undernutrition among children 6-59 months of age in East Africa.**

| Parameters | Models | | |
|---|---|---|---|
| | OLS | GWR | MGWR |
| Neighbors (Bandwidth) | – | 716 | – |
| Residual sum of squares | – | 3904.772 | 3850.050 |
| Effective number of parameters (trace(S)) | – | 360.962 | 320.195 |
| Sigma estimate | – | 0.829 | 0.821 |
| AICc | 15688.908 | 15274.916 | 15098.065 |
| R-Squared | 0.540 | 0.583 | 0.662 |
| Adjusted R-Squared | 0.539 | 0.557 | 0.643 |
| Log-likelihood | | −7252.350 | −7209.735 |

**Table 6. Summary Statistics for MGWR Parameter Estimates.**

| Variable | Mean | STD | Min | Median | Max | Bandwidth |
|---|---|---|---|---|---|---|
| Intercept | −0.025 | 0.216 | −0.579 | −0.012 | 0.526 | 161 |
| Proportion of maternal anemia | 0.151 | 0.108 | −0.079 | 0.157 | 0.414 | 346 |
| Proportion of no Vitamin A supplementation | 0.061 | 0.064 | −0.214 | 0.061 | 0.271 | 372 |
| Proportion of having Diarrhea | 0.025 | 0.073 | −0.147 | 0.017 | 0.304 | 406 |
| Proportion of having a Fever | 0.095 | 0.047 | 0.002 | 0.085 | 0.208 | 1008 |
| Proportion of tobacco user/smoker mothers | 0.014 | 0.101 | −0.228 | 0.08 | 0.28 | 913 |
| Proportion of the poorest households | 0.064 | 0.052 | −0.078 | 0.068 | 0.211 | 722 |
| Proportion of poorer households | 0.082 | 0.061 | −0.05 | 0.094 | 0.22 | 1252 |
| Proportion of no media exposure | 0.027 | 0.043 | −0.077 | 0.03 | 0.139 | 1428 |
| Proportion of birth interval <24 months, | 0.041 | 0.03 | −0.021 | 0.046 | 0.103 | 2039 |
| Proportion of maternal age 15–24 | 0.04 | 0.028 | −0.018 | 0.034 | 0.135 | 2243 |
| Proportion of no maternal education | 0.134 | 0.04 | 0.016 | 0.149 | 0.179 | 2315 |
| Proportion of multiple births | 0.065 | 0.018 | 0.035 | 0.071 | 0.096 | 4189 |
| Proportion of overweight/obese mothers | −0.065 | 0.002 | −0.068 | −0.066 | −0.061 | 6038 |
| Proportion of no health insurance | 0.056 | 0.003 | 0.05 | 0.056 | 0.061 | 6038 |
| Proportion of female sex | −0.047 | 0.008 | −0.058 | −0.048 | −0.024 | 5909 |
| Proportion of children aged 12–23 months | 0.06 | 0.003 | 0.054 | 0.06 | 0.064 | 6038 |
| Proportion of smaller/very small perceived birth size | 0.056 | 0.034 | −0.012 | 0.056 | 0.13 | 2163 |

(β = −0.079–0.0418) were found in Eastern Ethiopia, South-Central and North-Central Uganda, Eastern and Northern Tanzania, Western Zambia, and the southern parts of Mozambique and Zimbabwe (Fig 7).

The β coefficient for the proportion of children not receiving vitamin A supplementation in the six months preceding the survey showed both positive and negative associations with co-occurrence of anemia and undernutrition, ranging from −0.2114 to 0.2708. The highest β-values (0.1698–0.2708) were observed in eastern Uganda, central Rwanda, central Malawi, eastern-northern Zambia, and southeastern and northeastern Mozambique. In these regions, a unit increase in the proportion of children not receiving vitamin A supplementation corresponded to a higher prevalence of co-occurring anemia and undernutrition. Conversely, negative coefficients (as low as –0.2114) were found in northern Uganda, northeastern Ethiopia, central Burundi, northern to southeastern Tanzania, northeastern Zambia, and southeastern Zimbabwe, suggesting an inverse relationship between the lack of vitamin A supplementation and the co-occurrence of anemia and undernutrition in these areas (Fig 8).

The coefficient for recent history of diarrhea (occurrence within two weeks before the survey) exhibited both positive and negative associations with the outcome, ranging from −0.1473 to 0.3043, reflecting considerable spatial variability. The strongest positive associations (β = 0.1682 to 0.3043) were observed in eastern Zambia, eastern Tanzania, southern Mozambique, and northern Madagascar, where an increase in the proportion of children with recent diarrhea corresponded with higher rates of co-occurring anemia and undernutrition. In contrast, negative coefficients, as low as −0.1473, were found in most parts of Ethiopia and Tanzania, southeastern and northeastern Zimbabwe, eastern Uganda, and northern and eastern Malawi, indicating that a recent history of diarrhea was inversely associated with the co-occurrence of anemia and undernutrition in these regions (Fig 9).

The MGWR regression coefficient estimate of having a recent history of fever also demonstrated a positive and spatially varied relation with the co-occurrence of anemia and undernutrition in different Regions, ranging from 0.002 to 0.208. The red-colored cluster points (located in Southern Rwanda, most of Burundi, some parts of the Northwestern and Eastern to Southeastern part of Tanzania, Northeastern Mozambique, and most of the Northern part of Madagascar) imply

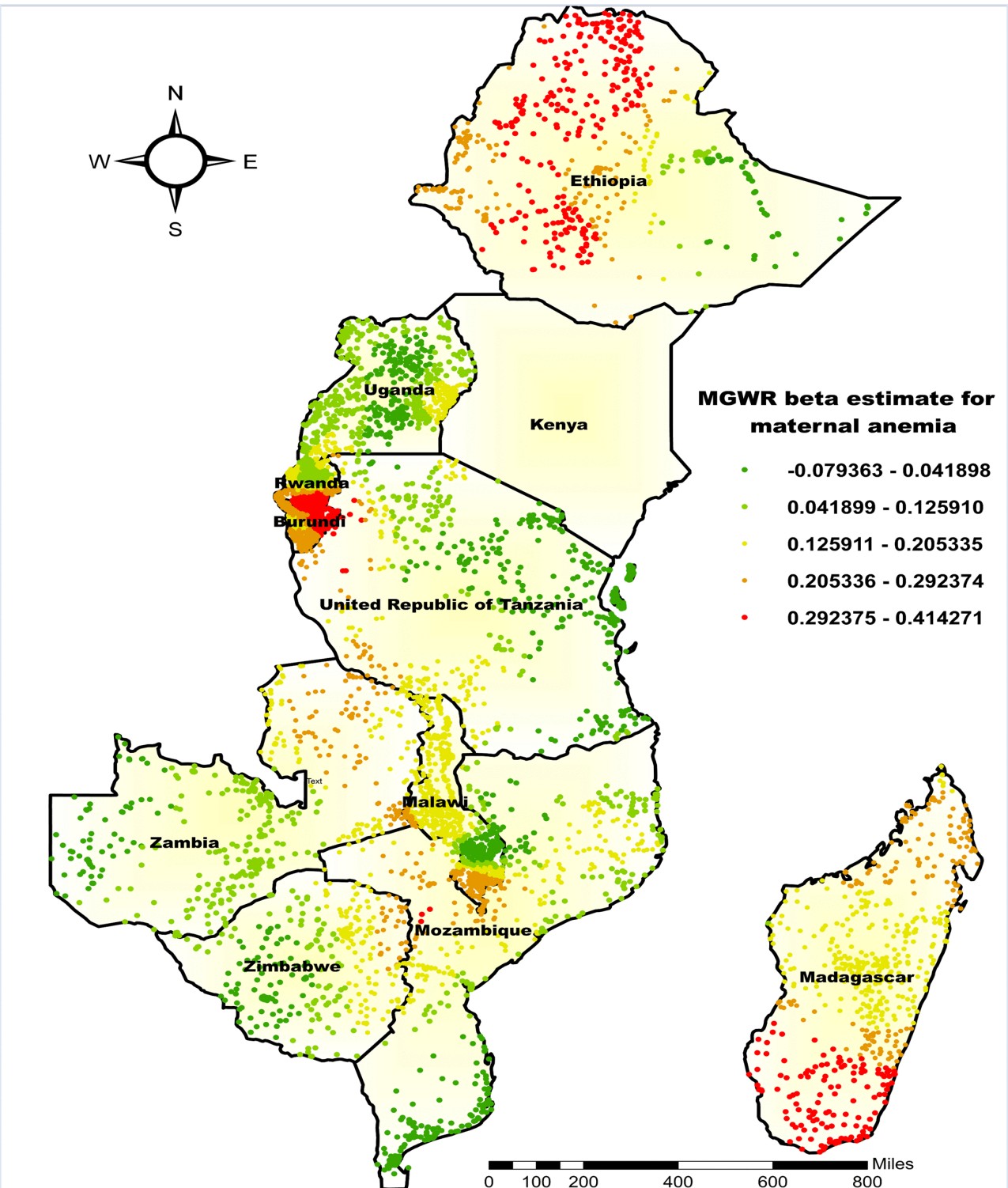

**Fig 7. MGWR coefficient estimates of being from an anemic mother for the co-occurrence of anemia and undernutrition among children 6-59 months in East Africa. Source**: Administrative boundary shapefile obtained from OpenAfrica (https://open.africa/dataset/africa-shapefiles). [**Note**: The authors utilized the shapefile solely as a basemap, performing all spatial processing, analysis, visualization, and modifications for illustrative and analytical purposes only.

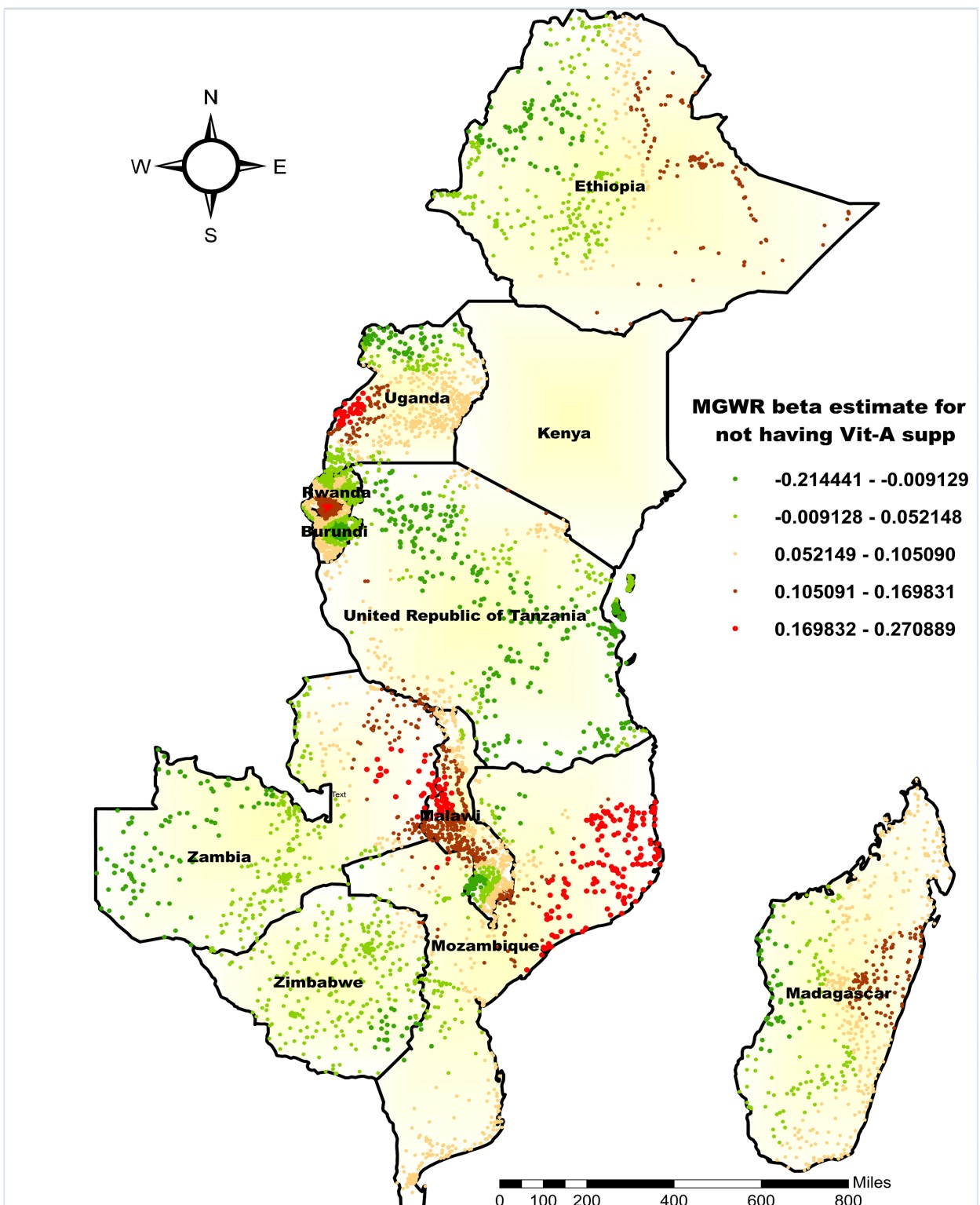

**Fig 8. MGWR coefficient estimates not having vitamin A supplementation for the co-occurrence of anemia and undernutrition among children 6-59 months in East Africa. Source**: Administrative boundary shapefile obtained from OpenAfrica (https://open.africa/dataset/africa-shapefiles). [**Note**: The authors utilized the shapefile solely as a basemap, performing all spatial processing, analysis, visualization, and modifications for illustrative and analytical purposes only].

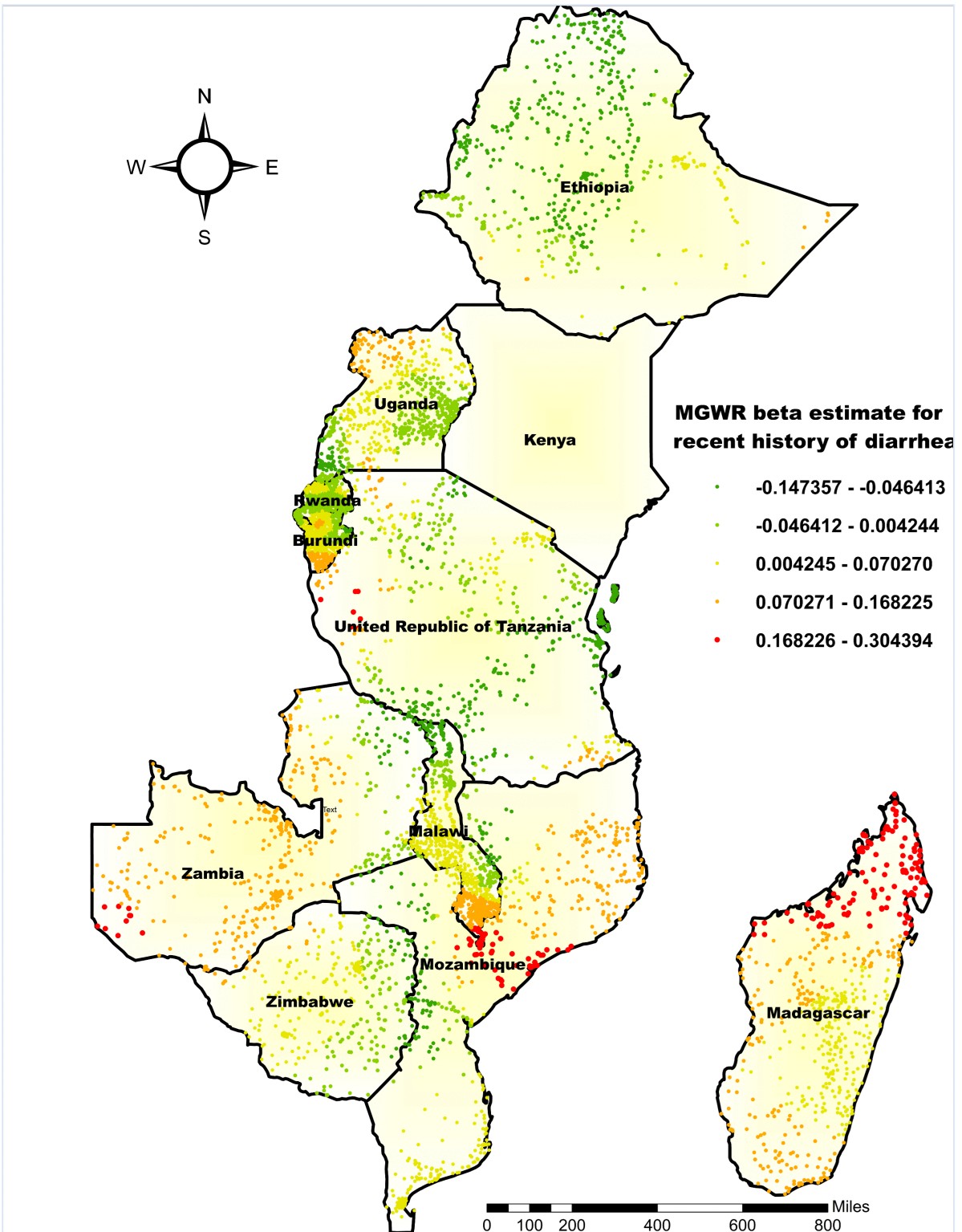

**Fig 9. MGWR coefficient estimates of having a recent history of diarrhea for the co-occurrence of anemia and undernutrition among children 6-59 months in East Africa. Source**: Administrative boundary shapefile obtained from OpenAfrica (https://open.africa/dataset/africa-shapefiles). [**Note**: The authors utilized the shapefile solely as a basemap, performing all spatial processing, analysis, visualization, and modifications for illustrative and analytical purposes only].

an area where there is a strong positive relationship. Conversely, in western Uganda and Southeastern Tanzania, North-eastern to Southeastern part of Zambia, as well as central to northeastern and southeastern Zimbabwe, the β-coefficient ranged from 0.002150 to 0.044566, suggesting a weaker but still significant positive association (Fig 10).

Regarding the coefficient estimate of maternal tobacco users/smokers, the β-estimate showed spatial variation with the highest positive β-coefficients, ranging from 0.1366 to 0.2795, concentrated in southern Tanzania and the northeast of Mozambique. While the lowest negative coefficient estimates, ranging from −0.220 to −0.1480, were observed in most of Malawi, Southern to Western Mozambique, and the Northeast part of Zimbabwe. In these regions, the proportion of tobacco user/smoker mothers was inversely associated with the co-occurrence of anemia and undernutrition prevalence (Fig 11).

Likewise, the MGWR beta coefficient estimates for being from the poorest household ranged from −0.078 to 0.211, with a mean value of 0.064, indicating spatial variability in its relationship with the co-occurrence of anemia and undernutrition across East Africa. The highest beta values, from 0.123 to 0.211, were observed in southeastern Uganda, northern Tanzania, some parts of eastern Burundi, southern Malawi, and central Mozambique. In these regions, a unit increase in the proportion of poorest households is associated with a 12.3 to 21.1 percentage point increase in the proportion of children experiencing co-occurrence of anemia and undernutrition. In contrast, the lowest beta coefficients, dropping to −0.078, were found in eastern Tanzania, most parts of Zambia, and northern Zimbabwe. In these areas, the impact of being from the poorest household on the co-occurrence of anemia and undernutrition prevalence was negative (Fig 12).

Furthermore, the MGWR coefficient estimates for belonging to a household with no media exposure regarding the co-occurrence of anemia and undernutrition among children aged 6–59 months in East Africa exhibited spatial variability, with beta coefficients ranging from −0.077 to 0.139 and an average value of 0.027. The highest β-values, from 0.087 to 0.139, were observed in northern Zimbabwe, southeastern Zambia, and the northern region of Mozambique. This suggests that in these areas, a greater proportion of households without media exposure was associated with a higher rate of co-occurring anemia and undernutrition. Specifically, a one-unit increase in the proportion of no media exposure corresponds to approximately a 0.087 to 0.139 point rise in the co-occurrence of anemia and undernutrition. Conversely, the lowest beta coefficients, dipping as low as −0.077, were recorded in nearly all regions of Uganda, southern Tanzania, and northern Madagascar. In these areas, the association between lack of media exposure and the co-occurrence of anemia and undernutrition was either weak or inverse (Fig 13).

The MGWR analysis revealed that a birth interval of less than 24 months had a weak overall positive association with the co-occurrence of anemia and undernutrition, with a mean beta coefficient of 0.041. However, the effect is not uniform across regions; some areas exhibit a negative association (minimum β = −0.021), while others show a stronger positive association (maximum β = 0.103) (Fig 14). The highest beta values, ranging from 0.076 to 0.103, were predominantly found in Ethiopia and Eastern Tanzania.

Similarly, being perceived as having a smaller or/and very small birth size was associated with anemia and undernutrition, with a mean beta coefficient of 0.056. This effect varied across regions: some areas showed a slight negative association (minimum β = −0.012), while others exhibited strong positive relationships (maximum β = 0.130). The highest beta values, between 0.102 and 0.130, were observed in Northern Malawi, Northern Zambia, and the southwestern region of Tanzania (Fig A in S2 File). Young maternal age (15−24 years) also showed an overall positive association with the co-occurrence of these conditions, with a mean beta coefficient of 0.04. The effect was inconsistent; some regions displayed negative associations (minimum β = −0.018), while others showed higher positive associations (maximum β = 0.135). The strongest associations, ranging from 0.082 to 0.135, were found in northern Zambia and Malawi, as well as central, northeastern, and western Tanzania (Fig B in S2 File).

Lack of maternal education exhibited a consistent positive association across all regions, with a mean β of 0.134 (range: 0.016–0.179). The strongest effects (β = 0.162–0.179) were found in western Ethiopia and most of Uganda (Fig C in S2 File). Likewise, multiple births were positively associated with the co-occurrence of anemia and undernutrition (mean

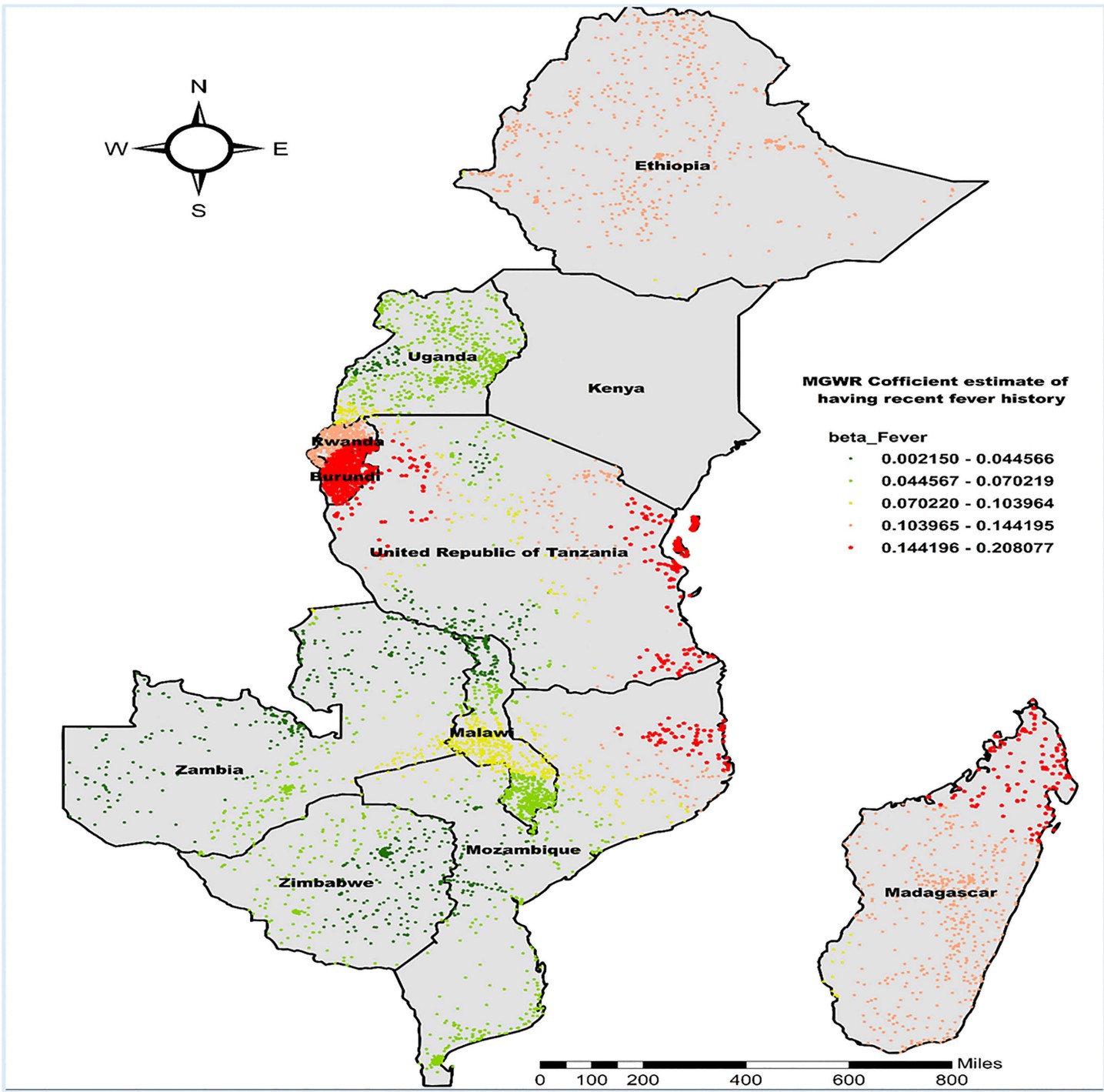

**Fig 10. MGWR coefficient estimates of having a recent history of fever for the co-occurrence of anemia and undernutrition among children 6-59 months in East Africa. Source**: Administrative boundary shapefile obtained from OpenAfrica (https://open.africa/dataset/africa-shapefiles). [**Note**: The authors utilized the shapefile solely as a basemap, performing all spatial processing, analysis, visualization, and modifications for illustrative and analytical purposes only].

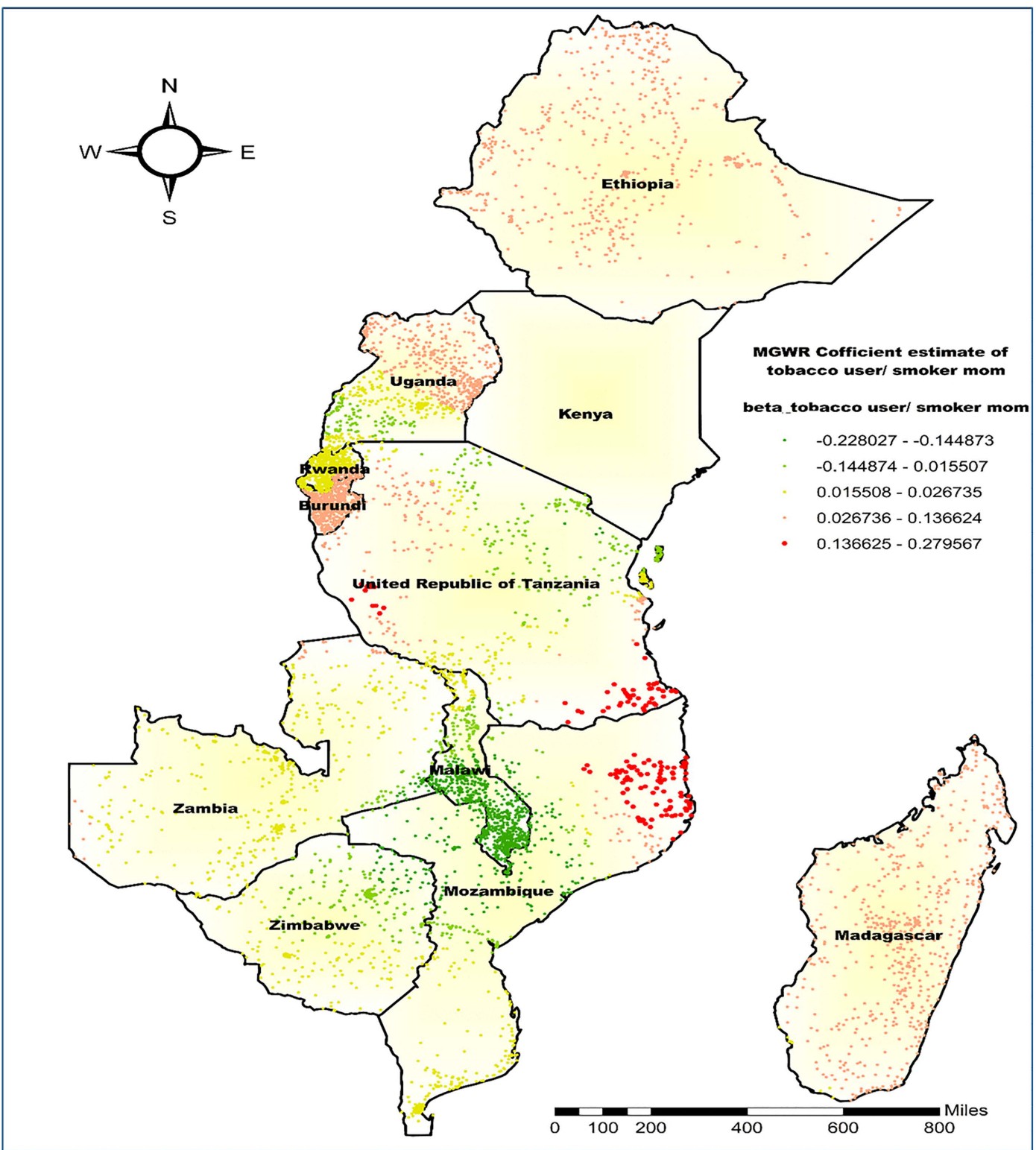

**Fig 11. MGWR coefficient estimates of being from tobacco users/smokers' mothers for the co-occurrence of anemia and undernutrition among children 6-59 months in East Africa. Source**: Administrative boundary shapefile obtained from OpenAfrica (https://open.africa/dataset/africa-shapefiles). [**Note**: The authors utilized the shapefile solely as a basemap, performing all spatial processing, analysis, visualization, and modifications for illustrative and analytical purposes only].

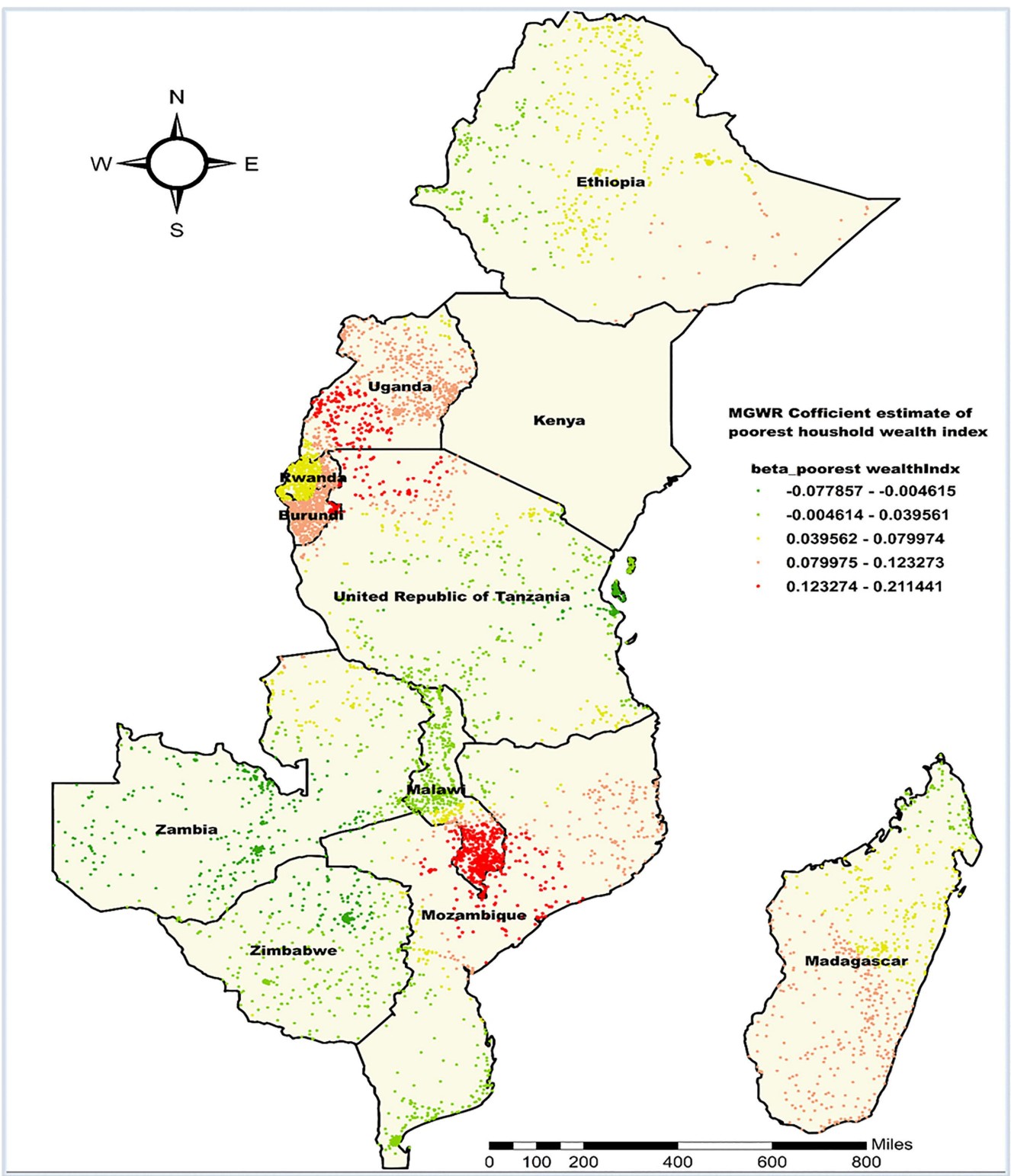

**Fig 12. MGWR coefficient estimates of being from the poorest households for the co-occurrence of anemia and undernutrition among children 6-59 months in East Africa. Source**: Administrative boundary shapefile obtained from OpenAfrica (https://open.africa/dataset/africa-shapefiles). [**Note**: The authors utilized the shapefile solely as a basemap, performing all spatial processing, analysis, visualization, and modifications for illustrative and analytical purposes only].

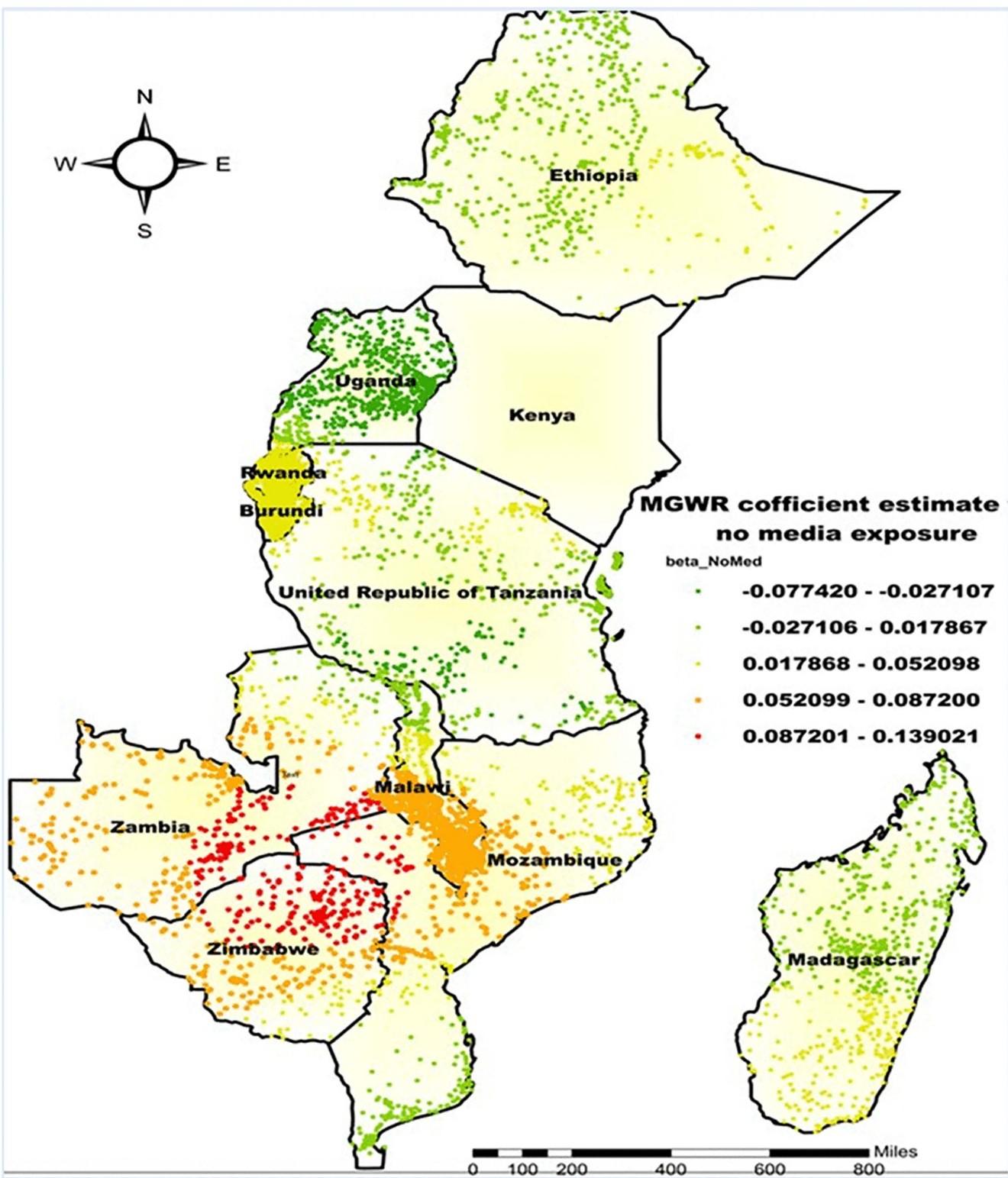

**Fig 13. MGWR coefficient estimates of being from a household with no media exposure for the co-occurrence of anemia and undernutrition among children 6-59 months in East Africa. Source**: Administrative boundary shapefile obtained from OpenAfrica (https://open.africa/dataset/afri-ca-shapefiles). [**Note**: The authors utilized the shapefile solely as a basemap, performing all spatial processing, analysis, visualization, and modifications for illustrative and analytical purposes only].

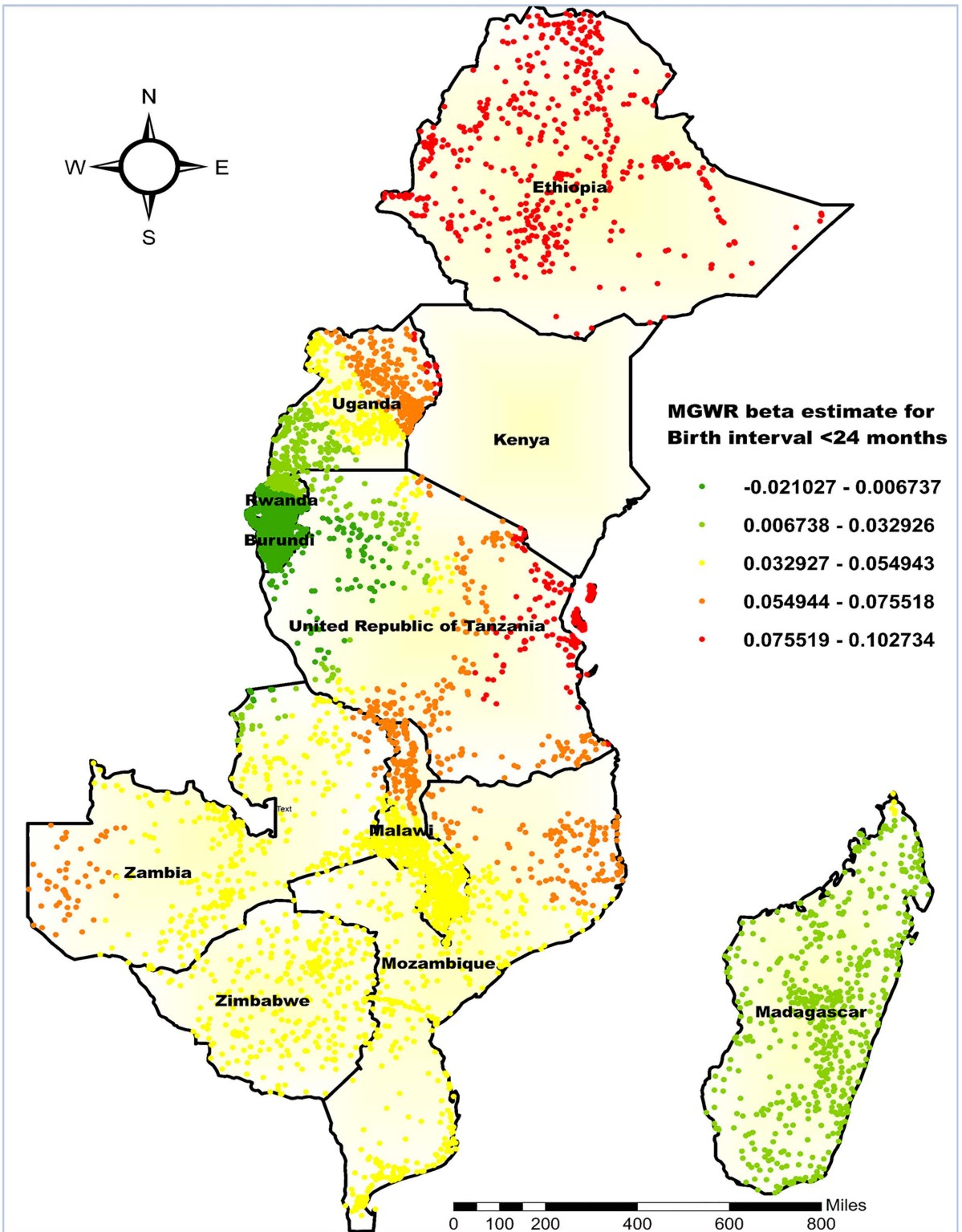

**Fig 14. MGWR coefficient estimates of birth interval of < 24 months for the co-occurrence of anemia and undernutrition among children 6-59 months in East Africa. Source**: Administrative boundary shapefile obtained from OpenAfrica (https://open.africa/dataset/africa-shapefiles). [**Note**: The authors utilized the shapefile solely as a basemap, performing all spatial processing, analysis, visualization, and modifications for illustrative and analytical purposes only].

β = 0.065), with coefficients ranging from 0.035 to 0.096. The highest values (β = 0.084–0.096) were found in southern Tanzania, northern Mozambique, northern Zambia, and much of Malawi (Fig D in S2 File).

Furthermore, the MGWR results indicated that certain predictors, being from an overweight mother, lack of health insurance, child age 12–23 months, and female sex, exhibited larger bandwidths, suggesting more global rather than localized effects. Children of overweight mothers showed a negative association with the co-occurrence of anemia and undernutrition (mean β = –0.065), indicating that areas with a higher proportion of overweight mothers tended to have lower rates of co-occurrence. Female sex (β = –0.047) also showed a negative association, indicating that areas with a higher proportion of female children had a lower prevalence of co-occurring anemia and undernutrition. In contrast, lack of health insurance was positively associated with the co-occurrence (β = 0.056), implying that regions with more children from uninsured households experienced higher co-occurrence rates. Similarly, children aged 12–23 months (β = 0.06) also showed a positive association, indicating that areas with a higher proportion of children in this age group also had a higher proportion of co-occurrence.

## Discussion

Anemia and undernutrition frequently co-exist, posing significant challenges to child growth, cognitive development, immune function, as well as overall survival, and are often rooted in intersecting biological, socioeconomic, and environmental determinants. By applying spatial analysis techniques, this research identified geographic hotspots where children face a higher risk and explored the predictors contributing to the co-occurrence of anemia and undernutrition. The study revealed a prevalence of 26.08% (95% CI: 25.70–26.46%) for the co-occurrence of anemia and undernutrition among children aged 6–59 months, highlighting the substantial public health burden of this dual form of malnutrition in the region. This prevalence varied across East African countries, with rates ranging from 13.36% in Zimbabwe to 40.63% in Burundi. This significant variation was further supported by the statistically significant Global Moran's I statistic, confirming a non-random spatial distribution. This result suggests that the burden of anemia and undernutrition is geographically clustered, likely influenced by socioeconomic/demographic and region-specific factors, including variation in health care services, variation in infectious diseases incidence, and feeding practices [32,33,40,41].

This study found that the co-occurrence of stunting and anemia was 23.48% (95% CI: 23.11–23.85%), underweight and anemia was 11.05% (10.78–11.33%), and wasting and anemia was 3.15% (95% CI: 3.0–3.3%) among children 6–59 months in East Africa. The co-occurrence of stunting and anemia observed in this study is lower than that reported in a nationally representative study from Ethiopia, which found a prevalence of 24.4% [32]. However, it is higher than the estimates from a study conducted across 43 countries (21.5%) [42] and a meta-analysis involving 46 least-developed countries, which reported a prevalence of 19.47% [43]. Regarding the co-occurrence of wasting and anemia, the finding in this study is lower compared to findings in Ethiopia (6.3%) [44] and also lower than the estimate from a meta-analysis study (5.44%) [43].

The observed differences in prevalence may be attributed to several methodological and contextual factors. For example, the first study from 43 low- and middle-income countries [42], encompassing 193,065 children, while our study included 50,760 children, and these differences in sample sizes and study scopes likely contribute to the variation in reported prevalence rates. While our study focused on a cross-sectional survey of East African countries, the meta-analysis involving 46 least-developed countries and other multi-country analyses [43] provides a broader perspective on the issue across a wide range of low- and middle-income countries. Such differences in geographical coverage can lead to variations in findings due to diverse environmental, cultural, and policy contexts. Additionally, disparities in the socio-economic profiles of the populations studied, including factors such as income level, education, access to healthcare, and nutritional practices, could contribute to the variation in reported prevalence rates [32,33,41].

The hotspot analysis revealed significant spatial variations in the co-occurrence of anemia and undernutrition, with high-risk areas concentrated in Northeast Ethiopia, Northern and Northwest Uganda, most of Burundi, Northeast

Tanzania, Northern Zambia, Central to Southern Malawi, much of Eastern Mozambique, and Southern Madagascar. In contrast, low-prevalence areas were found in Central Ethiopia, Southeast and Central to Eastern Uganda, most parts of Zimbabwe, and the Southern region of Mozambique. Similarly, the ordinary kriging interpolation predictions for unsampled areas also confirmed that the highest burden was primarily located in eastern and southern Ethiopia, northern Uganda, northeastern Burundi, north of Zambia, the southern to eastern regions of Mozambique, and northern Madagascar. This spatial disparity may stem from differences in healthcare access, the availability of nutritional interventions [43,45], socio-economic conditions [46], as well as cultural and geographic factors [45]. For instance, in certain regions of Uganda and Ethiopia, limited healthcare infrastructure, low maternal education, and food insecurity were high and may contribute to the higher prevalence, while areas with better access to health services and improved socioeconomic conditions report lower prevalence [47,48]. Furthermore, variations in dietary practices [43], disease burden and health policies across countries may explain the uneven distribution observed. These findings emphasize the importance of geographically targeted, evidence-based interventions to reduce the burden in high-risk areas

The MGWR analysis showed a strong positive link between maternal anemia and the co-occurrence of anemia and undernutrition in specific regions of Ethiopia, Burundi, and Madagascar. This may be due to anemic mothers having low iron stores during pregnancy and lactation, which can result in poor iron transfer to their children and raise their risk of both conditions [49]. Additionally, anemic mothers may experience fatigue and poor health, reducing their ability to provide optimal childcare, including breastfeeding and complementary feeding practices [50]. Most importantly, maternal anemia may reflect broader contextual factors such as household food insecurity, suboptimal maternal dietary practices, and limited access to nutrient-rich foods. These factors can directly influence child feeding practices and contribute to the co-occurrence of anemia and undernutrition, highlighting the complex interplay between maternal and child nutritional status within the household and socio-economic context [51,52] The association of maternal anemia with the co-occurrence of anemia with undernutrition aligns with previous studies that have emphasized the role of maternal anemia in affecting child nutrition [10,32]. Discrepancies in the strength of the association across the study area could be due to variations in other local factors, including healthcare access and cultural practices related to diet.

Similarly, the MGWR analysis identified that not receiving Vitamin A supplementation in the six months preceding the survey had strong positive associations with the co-occurrence of anemia and undernutrition in specific regions, including northeastern Ethiopia, eastern Uganda, northern Burundi, central Malawi, eastern-northern Zambia, southeastern and northeastern Mozambique, and parts of eastern Madagascar. These results support existing evidence of widespread Vitamin A deficiency in the region and emphasize the need for targeted interventions to improve supplementation and address regional disparities [43,53]. Vitamin A supplementation helps prevent anemia and undernutrition by facilitating iron mobilization, enhancing red blood cell production, improving nutrient absorption, and supporting growth and immune function. However, in some clusters, a counterintuitive inverse association was observed, where lack of recent supplementation was associated with lower co-occurrence. This finding should not be interpreted as a causal effect of withholding vitamin A, but rather as a reflection of place-specific healthcare delivery patterns and residual confounding. In many settings, vitamin A supplementation is provided through integrated child health campaigns that target nutritionally vulnerable children and bundle multiple services [54,55]. As a result, supplemented children may represent populations with a higher underlying burden of illness and nutritional risk, producing an apparent inverse association due to programmatic targeting and indication bias. The observed regional variability likely reflects differences in underlying deficiency burden, sociodemographic conditions, and the extent to which supplementation programs are effectively integrated into local health services, emphasizing the need for geographically targeted interventions [56,57].

The findings from the MGWR analysis indicated that a recent history of diarrhea showed spatial variation in predicting the co-occurrence of anemia and undernutrition, with the strongest predictors observed in eastern Zambia, Tanzania, southern Mozambique, and northern Madagascar, supporting existing evidence that diarrheal diseases impair children's nutritional status through nutrient malabsorption and losses [58] and increasing the risk of anemia and undernutrition

[59,60]. However, a few regions exhibited negative local coefficients, indicating an inverse association. This does not contradict the established link between diarrhea and malnutrition but highlights how local contextual factors can modify population-level relationships. In areas with better health service coverage, children experiencing diarrhea may receive timely treatment, rehydration, micronutrient supplementation, nutritional counseling, or enhanced caregiving, which could temporarily reduce the risk of co-occurring anemia and undernutrition [58,61]. Moreover, spatial heterogeneity reflects differences in water and sanitation access, enteric pathogen exposure, and healthcare availability; for example, hotspots with limited safe water and poor sanitation likely experience higher diarrheal burden, increasing malnutrition risk [62]. Hence, these findings highlighted the importance of considering spatial heterogeneity and caution against interpreting the inverse association as a general protective effect.

The identification of high beta coefficient values of recent history of fever in the MGWR analysis, particularly in Southern Rwanda, most of Burundi, and parts of Tanzania, Mozambique, and Madagascar, exhibited strong positive relationships between recent fever episodes and the simultaneous presence of anemia and undernutrition. Fever in children often indicates underlying infections, which can impair nutrient absorption and increase metabolic demands, thereby exacerbating nutritional deficiencies and contributing to anemia [16]. Febrile illnesses like malaria are common in regions such as Burundi, Madagascar, and Mozambique, likely contributing to the strong link between fever history and the co-occurrence of anemia and undernutrition there [63,64]. This interpretation aligns with previous studies that have identified a history of fever or acute infections as a significant contributor to the co-occurrence of anemia and undernutrition in under-five children, as infections can exacerbate nutritional deficiencies [10,65].

Regarding maternal tobacco use, the highest beta coefficients, concentrated in southern Tanzania and northeast Mozambique, indicate a strong positive association with the co-occurrence of anemia and undernutrition. This suggests that maternal smoking during pregnancy and postpartum increases children's nutritional vulnerability, consistent with research showing its harmful effects on birth weight, nutrient absorption, and growth that can continue into early childhood [66,67]. Current maternal tobacco use may predispose infants to anemia, thinness, and growth problems [66,68]. This connection is further supported by studies in Indonesia and Bangladesh, which have associated tobacco use with increased risks of underweight and wasting, indicating a consistent pattern across diverse geographical contexts [67,69]. Regional disparities may result from confounding factors such as cultural practices, socioeconomic differences, and varying tobacco use among women, with countries like Tanzania having higher tobacco exposure due to its status as a major tobacco producer [70].

The findings from this study using the MGWR model also highlighted that being from a household with no media exposure had a positive influence on the co-occurrence of anemia and undernutrition. The strongest positive associations were observed in Northern Zimbabwe, Southeast Zambia, and Northern Mozambique. From previous studies, it has been reported that a lack of media access was significantly associated [71]. This might be due to the presence of some factors, including a lack of access to information and awareness about proper nutrition, hygiene, and healthcare practices among households with limited media exposure [72]. Parental awareness and access to information were essential for successful child nutrition interventions, with media access increasing exposure to health messages and healthcare utilization [73].

In addition, the highest positive associations of being from the poorest household using MGWR coefficient estimates were observed in southeastern Uganda, northern Tanzania, parts of eastern Burundi, southern Malawi, and central Mozambique. Previous studies have consistently shown that children from the poorest households are at a higher risk of experiencing both anemia and undernutrition indices [74–76]. This may be due to the compounded effects of limited access to nutritious food, inadequate sanitation, and restricted healthcare services that are characteristic of the poorest households in these regions. Specifically, in these areas, food insecurity may be more pronounced, leading to undernutrition and anemia [77]. The high associations in these specific regions highlight the urgent need for targeted poverty reduction and nutrition-sensitive interventions to address the dual burden of malnutrition in vulnerable populations.

This study further revealed that certain predictors, like, maternal overweight status, lack of health insurance, child age (12–23 months), and female sex, exhibited global effects on the co-occurrence of anemia and undernutrition rather than

strong localized variations. Maternal overweight status was negatively associated with anemia and undernutrition, aligning with previous studies suggesting that children of overweight mothers may have better nutritional status due to food security, improved maternal nutrient stores, and dietary practices [78,79]. Conversely, the lack of health insurance showed a positive association with anemia and undernutrition, consistent with findings indicating that uninsured households face barriers to healthcare access, leading to lower utilization of preventive and curative services for childhood illnesses and malnutrition [10,80].

Moreover, being aged 12–23 months was positively associated with anemia and undernutrition, which may be attributed to the critical nature of the weaning period, where children in this age group often experience inadequate dietary diversity and increased exposure to infections [10], emphasizing the need for age-specific nutritional interventions. This is in line with previous research highlighting that children in this age bracket are at heightened risk for malnutrition due to suboptimal feeding practices and high rates of morbidity. In contrast, female children had a negative association with anemia and undernutrition, which is consistent with studies. This might be due to biological advantages in metabolism or cultural differences in feeding practices, suggesting that boys are biologically more vulnerable to infections and growth faltering during infancy and early childhood [81].

### Strengths and limitations of the study

This study possesses some notable strengths. Firstly, the use of DHS data ensures the representativeness and reliability of the findings, offering insights into the geographic variability of the problem. Second, the use of advanced spatial analysis models allowed for the exploration of spatially varying relationships between predictors and the outcome, and identifying specific hotspot areas and spatial clusters provides a good basis for prioritizing interventions and allocating resources effectively. Finally, integrating spatial analytical methods strengthens the ability to identify hotspot regions where targeted interventions are most needed, supporting evidence-based public health planning.

Despite its strengths, this study is subject to several limitations. Firstly, the cross-sectional nature of the DHS data precludes the establishment of causal relationships between predictors and the co-occurrence of anemia and undernutrition. Second, while the study incorporates various maternal, child, and household factors, other potential determinants, such as household food insecurity and infectious disease prevalence, were not fully accounted for due to data limitations. While minimum dietary diversity of a child was included as a covariate, the lack of a direct household food insecurity measure in the DHS data remains a limitation. Third, since biomarker collection was performed on selected households, children without anemia and anthropometric data were excluded, potentially introducing selection bias. Although this approach was necessary to ensure valid measurement of co-occurrence, the findings may not fully represent all children in the survey population. In addition, since the data is based on self-reporting, social desirability bias and recall bias might be present

### Conclusion

This study examined the spatial distribution and predictors of the co-occurrence of anemia and undernutrition among children aged 6–59 months in East Africa using recent DHS data. The findings revealed the significant geographic disparities with non-random spatial distribution. Hotspot areas were identified in northeast Ethiopia, northern and northwest Uganda, most parts of Burundi, northeast Tanzania, northern Zambia, central to southern Malawi, eastern Mozambique, and southern Madagascar. Spatial regression analysis revealed that maternal anemia, tobacco use, lack of education, poor household wealth, lack of health insurance, and recent child diarrhea and fever are key predictors of anemia and undernutrition, with their effects varying by region. These findings highlight the need for region-specific interventions, such as improving maternal nutrition, promoting child feeding practices, and addressing household socioeconomic challenges, strategies that have been shown in prior studies to reduce child anemia and undernutrition. Targeted public health programs focusing on identified hotspots, alongside future longitudinal research to explore additional confounders, are recommended to effectively mitigate the burden in East Africa.

## Supporting information

**S1 File. Supplementary spatial distribution analysis results.** This file contains one table and multiple figures related to the spatial distribution and analysis of the co-occurrence of anemia and undernutrition among children aged 6–59 months in East Africa.S1 Fig A. Spatial distribution of co-occurrence of anemia and undernutrition. S1 Fig B. *Incremental autocorrelation for the co-occurrence of anemia and undernutrition.* S1 Fig C. *Ordinary Kriging interpolation for the co-occurrence of anemia and undernutrition.* S1 Table A. *Spatial SaTscan analysis results for the co-occurrence of anemia and undernutrition.*
(DOCX)

**S2 File. Supplementary spatial regression analysis results.** This file contains additional outputs from the spatial regression analyses for the co-occurrence of anemia and undernutrition among children aged 6–59 months in East Africa. S2 Table A: Summary statistics of the estimated coefficients for local terms in GWR model. S2 Fig A: MGWR coefficient estimates for children perceived as having smaller/very small birth size for the co-occurrence of anemia and undernutrition. S2 Fig B: MGWR coefficient estimates for maternal age 15–24 years for the co-occurrence of anemia and undernutrition. S2 Fig C: MGWR coefficient estimates of mothers with no formal education for the co-occurrence of anemia and undernutrition. S2 Fig D: MGWR coefficient estimates of multiple birth for the co-occurrence of anemia and undernutrition.
(DOCX)

**S3 File. Raw data used for statistical analyses.** This file contains anonymized minimal raw data used for descriptive, regression, and spatial analyses of the co-occurrence of anemia and undernutrition among children aged 6–59 months in East Africa.
(CSV)

## Author contributions

**Conceptualization:** Altaseb Beyene Kassaw.

**Data curation:** Altaseb Beyene Kassaw.

**Formal analysis:** Altaseb Beyene Kassaw.

**Investigation:** Anissa Mohammed, Amare Muche.

**Methodology:** Altaseb Beyene Kassaw, Asressie Molla.

**Software:** Altaseb Beyene Kassaw.

**Supervision:** Asressie Molla.

**Validation:** Anissa Mohammed, Amare Muche, Asressie Molla.

**Visualization:** Altaseb Beyene Kassaw.

**Writing – original draft:** Altaseb Beyene Kassaw.

**Writing – review & editing:** Anissa Mohammed, Amare Muche, Asressie Molla.

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
