## [Decision Letter · Decision Letter 0]

2 Jan 2026

Dear Dr. Kassaw,

Thank you for submitting your manuscript to PLOS ONE. We have now received the reviewers' comments on your submission. We invite you to submit a revised version of the manuscript that addresses the points raised during the review process.

We look forward to receiving your revised manuscript.

Kind regards,

Neetu Choudhary, PhD

Academic Editor

PLOS One

Journal Requirements:

4. We note that Figure(s) 1, 2, 3, 4, S2 in your submission contain copyrighted images. All PLOS content is published under the Creative Commons Attribution License (CC BY 4.0), which means that the manuscript, images, and Supporting Information files will be freely available online, and any third party is permitted to access, download, copy, distribute, and use these materials in any way, even commercially, with proper attribution. For more information, see our copyright guidelines: http://journals.plos.org/plosone/s/licenses-and-copyright.

a. You may seek permission from the original copyright holder of Figure(s) 1, 2, 3, 4, to publish the content specifically under the CC BY 4.0 license.

5. We note that Figure(s) 5, 6, 7, 8, 9, 10, 11, 12, 13, 14, Supplementary file 1, Supplementary file 2 in your submission contain [map/satellite] images which may be copyrighted. All PLOS content is published under the Creative Commons Attribution License (CC BY 4.0), which means that the manuscript, images, and Supporting Information files will be freely available online, and any third party is permitted to access, download, copy, distribute, and use these materials in any way, even commercially, with proper attribution. For these reasons, we cannot publish previously copyrighted maps or satellite images created using proprietary data, such as Google software (Google Maps, Street View, and Earth). For more information, see our copyright guidelines: http://journals.plos.org/plosone/s/licenses-and-copyright.

a. You may seek permission from the original copyright holder of Figure(s) 5, 6, 7, 8, 9, 10, 11, 12, 13, 14, Supplementary file 1, Supplementary file 2 to publish the content specifically under the CC BY 4.0 license.

6. We are unable to open your Supporting Information file [Supplementary file 3. Data set.dta]. Please kindly revise as necessary and re-upload.

7. We note that there is identifying data in the Supporting Information file <Supplementary file 4. GPS data.xls>. Due to the inclusion of these potentially identifying data, we have removed this file from your file inventory. Prior to sharing human research participant data, authors should consult with an ethics committee to ensure data are shared in accordance with participant consent and all applicable local laws.

-Location data

Reviewers' comments:

Reviewer's Responses to Questions

**Comments to the Author**

1. Is the manuscript technically sound, and do the data support the conclusions?

Reviewer #1: Yes

Reviewer #2: Yes

2. Has the statistical analysis been performed appropriately and rigorously?

Reviewer #1: Yes

Reviewer #2: Yes

3. Have the authors made all data underlying the findings in their manuscript fully available?

Reviewer #1: Yes

Reviewer #2: Yes

4. Is the manuscript presented in an intelligible fashion and written in standard English?

Reviewer #1: Yes

Reviewer #2: Yes

Reviewer #1: Spatial Distribution and Factors Associated with Co-occurrence of Anemia and Undernutrition among Children Aged 6-59 Months in East Africa. Good work done in the analyses structures. However, check the following observation

In the abstract, how was the co-occurrence measured?

Fig 1

1 Check the numbers: "... countries have DHS"

2. More than 50% participants were excluded. This is much. Why have you not impute missing values?

Eligibility

Inclusion context: What is the measure for 'most recent DHS?'

Child-related factors. You have 'child twins status'. What of 'multiple birth'?

Data quality control

Lns 222-224. 'with appropriate methods of imputation'. What do you mean? Did you perform any imputation for the data?

Recoding the outcome variable Lns 267-268. Why have you decided to recode the outcome variable into binary rather than used as multinomial?

How did you access the multicollinearity and what were the characteristics?

Ln 299 Check the full meaning of MGWR

Ln 352. This may be controversial? Why have you considered Zimbabwe as part of East Africa countries. Give a short explanation for your classification

Fig 3 Did you attempted checking if there are interactions between the indicators of undernutrition, and for possible three-way interactions

Limitations

You said food insecurity was not considered. Why? Some research used minimum dietary diversity as proxy for food insecurity. Why have you not used this?

Reviewer #2: Comments to the authors:

Line 346 table 2,

Please carefully recheck and revise the percentage calculations in the tables, as several variables do not sum to 100%, including maternal literacy, maternal anemia, drinking water, sanitary facilities, birth order, and cluster altitude.

You have specified the anemia cutoff for children under five; however, the cutoff used to define maternal anemia has not been clearly stated.

Line 501-505

The observed inverse association between the lack of vitamin A supplementation and the co-occurrence of anemia and undernutrition should be clearly explained and contextualized in the Discussion section, with reference to existing evidence.

Line 512-516

The inverse association between recent diarrhea and the co-occurrence of anemia and undernutrition is contrary to existing evidence. This finding should be critically explained and discussed in the Discussion section.

Line 679-684

While the explanation focusing on low maternal iron stores and reduced caregiving capacity is valid, the discussion should also consider broader contextual factors. Maternal anemia may reflect underlying household food insecurity and suboptimal maternal dietary practices, which can directly influence child feeding practices and contribute to the co-occurrence of anemia and undernutrition in children. These pathways should be acknowledged and discussed.

Line 793 conclusion

The recommended strategies in the Conclusion should be supported by relevant evidence or successful intervention examples in the Discussion.

**Do you want your identity to be public for this peer review?** For information about this choice, including consent withdrawal, please see our Privacy Policy

Reviewer #1: **Yes:** Phillips Obasohan

Reviewer #2: No

---

## [Author Response · Author response to Decision Letter 1]

12 Jan 2026

Dear Editor and Reviewers,

Thank you very much for your support and the opportunity to resubmit our manuscript entitled "Spatial Distribution and Factors Associated with Co-occurrence of Anemia and Undernutrition among Children Aged 6-59 Months in East Africa"(Manuscript ID: PONE-D-25-34030). We sincerely appreciate the detailed comments and helpful suggestions provided. We have thoroughly addressed all points raised and believe the manuscript has been significantly improved to meet the journal’s standards. We welcome any additional feedback you may have.

We have revised the manuscript accordingly and submitted the required documents, including:

• A marked-up copy of the revised manuscript highlighting all changes.

• A clean version of the revised manuscript without tracked changes.

• A point-by-point response to all comments

In addition, we have ensured full compliance with PLOS ONE guidelines, including manuscript formatting and file naming, placement of the ethics statement within the Methods section, verification of author affiliations, resolution of copyright issues for all figures and maps in accordance with the CC BY 4.0 license, and inclusion of complete captions for all Supporting Information files.

Below, you will find our point-by-point responses to the comments. The comments are in italicized black, followed by our responses in blue. All changes made to the original manuscript are highlighted in red within the marked-up copy.

Thank you once again for your guidance and thoughtful consideration.

Kind regards,

Altaseb Beyene Kassaw

On behalf of all the authors.

Response to Journal Requirements

Comment 1. Please ensure that your manuscript meets PLOS ONE's style requirements, including those for file naming.

Response: We have revised the manuscript to fully comply with PLOS ONE formatting and style requirements.

Comment 2: Your ethics statement should only appear in the Methods section of your manuscript. If your ethics statement is written in any section besides the Methods, please delete it from any other section.

Response: We have re-checked that the ethics statement appears only in the Methods section.

Comment 3: Please amend your list of authors on the manuscript to ensure that each author is linked to an affiliation.

Response: The author list has been amended so that each author is clearly linked to their respective institutional affiliation.

Comment 4: Copyrighted figures (Figures 1, 2, 3, 4, and S2)

Response: Thank you for raising this concern. We confirm that Figures 1, 2, 3, and 4 are entirely original works created by the authors. All figures were generated by us using our own data and analytical outputs and do not reproduce, adapt, or derive from any copyrighted images.

To further ensure full compliance with PLOS ONE’s CC BY 4.0 licensing requirements, we have updated the figures (e.g., color schemes and visual styling) to clearly distinguish them as our original creations. If required, we are happy to provide the STATA code used to generate some of the figures as verification of originality.

If there are specific images that the editorial team considers similar to our figures, we would appreciate it if these could be shared with us so that we can address any remaining concerns. Accordingly, no third-party copyrighted material is included in the revised figures, and no permissions are required.

Regarding Supplementary file 2, it actually contains maps created by the authors using ArcGIS based on open-access basemap and shapefile data. The basemap layers used in ArcGIS were obtained from the OpenAfrica website, which provides public-domain map data suitable for reuse and redistribution. Further explanation is included under “Response to Comment 5”.

Comment 5: (Map and Satellite image licensing, (Figures 5–14; Supplementary Files 1 and 2). We note that Figure(s) 5, 6, 7, 8, 9, 10, 11, 12, 13, 14, Supplementary file 1, Supplementary file 2 in your submission contain [map/satellite] images which may be copyrighted………..

Response: Thank you for raising this important issue.

We would like to clarify that all map-based figures (Figures 5–14) and Supporting Information files (Figures in Supplementary Files 1 and 2) were created by the authors using openly licensed shapefiles as basemap layer, which were obtained from the OpenAfrica platform (https://open.africa/dataset/africa-shapefiles). These shapefiles were used exclusively as basemap layers, and all subsequent spatial processing, statistical analyses, and visualizations were independently conducted by the authors. No proprietary software or copyrighted map imagery (e.g., Google Maps or Google Earth) was used in the preparation of these figures.

We have now updated the figures and revised the figure captions to acknowledge the OpenAfrica data source and its direct link to the data source in each corresponding figure legend. As all figures were generated using open-access data and author-created visualizations, we respectfully submit without additional copyright permission.

Comment 6: “We are unable to open your Supporting Information file [Supplementary file 3. Data set.dta]. Please kindly revise as necessary and re-upload”.

Response: Supplementary File 3 has now been re-uploaded in CSV format.

Comment 7: “We note that there is identifying data in the Supporting Information file <Supplementary file 4. GPS data.xls>. Due to the inclusion of these potentially identifying data, we have removed this file from your file inventory………………….”

Response: We acknowledge that Supplementary File 4 (GPS data.xls) contained potentially identifying information, and to ensure compliance with PLOS ONE’s policies, this file has been removed from the submission.

Comment 8: “Please include captions for your Supporting Information files at the end of your manuscript, and update any in-text citations to match accordingly…….”

Response: We have now added captions for all Supporting Information files at the end of the manuscript, following the PLOS ONE guidelines. In-text citations to the Supporting Information files have also been updated.

Responses to Reviewers’ Comments

Reviewer 1:

Comment 1: [In the abstract, how was the co-occurrence measured?]

Response: Thank you for this important comment. We have revised the abstract to clearly specify how the co-occurrence of anemia and undernutrition was measured.

Comment 2: [Fig 1: (1) Check the numbers: "... countries have DHS". (2). More than 50% participants were excluded. This is much. Why have you not impute missing values?].

Response: Thank you for these important observations.

We carefully rechecked the number of countries with available DHS data and confirmed that the number presented in Figure 1 is correct.

The large proportion of excluded participants was primarily due to the DHS survey design. Biomarker data, including anemia testing, are not collected from all households in a DHS survey. Instead, from all households selected for the main DHS survey, only a subsample is selected for biomarker measurements. Consequently, hemoglobin data required to define anemia were unavailable for a substantial number of children.

Given that missingness arose from survey design rather than random omission, we did not perform imputation. Moreover, the missingness was primarily related to outcome variables rather than covariates, and imputation of outcome data may introduce additional bias. Therefore, a complete case analysis was conducted to preserve the validity of outcome classification.

Comment 3:[Child-related factors. You have 'child twins status'. What of 'multiple birth'?]

Response: Thank you for pointing out. The variable “child twin status” was coded as single birth or multiple birth (≥2), thereby capturing all multiple births, including twins and higher-order multiples.

Comment 4: Data quality control: [ Lns 222-224. 'with appropriate methods of imputation'. What do you mean? Did you perform any imputation for the data?].

Response:. Thank you for this comment. We would like to clarify that no imputation was performed for the key outcome variables (hemoglobin levels and anthropometric indicators), as their missingness was primarily due to the DHS survey design. Accordingly, a complete case analysis was conducted, including only children with available hemoglobin and anthropometric data. However, appropriate imputation was applied for missing values of a few covariates (including perceived size of child at birth) for the analysis.

[Recoding the outcome variable Lns 267-268. Why have you decided to recode the outcome variable into binary rather than used as multinomial?].

Response: The outcome variable was recoded into a binary variable (co-occurrence vs. no co-occurrence) to focus specifically on children experiencing both anemia and undernutrition simultaneously. This approach simplifies interpretation, allows clear assessment of associated factors, and aligns with the study’s primary objective of identifying predictors of co-occurrence.

We believe using a multinomial outcome would have required separating each combination of anemia severity level and undernutrition types, creating multiple small categories and complicating the modeling. However, if the reviewer believes that a multinomial approach would provide stronger statistical insights, we are open to exploring this in a future study.

Comment 5:

[“How did you access the multicollinearity and what were the characteristics?” ]

Response: Multicollinearity was assessed during exploratory regression analysis using the Variance Inflation Factor (VIF). There was no evidence of multicollinearity among the included set of explanatory variables, as all VIF values were below 7.5. Only variables with VIF below 7.5 were included in the final analysis.

Comment 6: [Ln 299 Check the full meaning of MGWR]

Response: The full meaning of MGWR has been clarified in the manuscript as Multiscale Geographically Weighted Regression (MGWR).

Comment 7: [Ln 352. This may be controversial? Why have you considered Zimbabwe as part of East Africa countries. Give a short explanation for your classification].

Response: Thank you for raising this point. Yes, of course, Zimbabwe is not classified as part of East Africa in standard geographical or regional frameworks; it is commonly described as a country in Southern Africa. In this study however, it was classified as part of East Africa based on the United Nations geoscheme for Africa, which groups countries into regional classifications for statistical and analytical purposes. This classification has been widely used in multi-country demographic and health analyses to ensure regional comparability and data availability.

Sample Sources:

1). https://unstats.un.org/unsd/methodology/m49/

2). https://worldpopulationreview.com/country-rankings/east-african-countries

3). https://www.worldatlas.com/geography/east-african-countries.html

Comment 8: [“Fig 3 Did you attempted checking if there are interactions between the indicators of undernutrition, and for possible three-way interactions”].

Response: Thank you for this thoughtful question. We did not explicitly model interaction terms among the individual indicators of undernutrition (stunting, wasting, and underweight), nor three-way interactions, because our outcome variable was defined as the co-occurrence of anemia and any form of undernutrition, rather than the joint occurrence of specific undernutrition indicators. Stunting, wasting, and underweight were therefore combined into a single composite indicator of undernutrition for the primary analysis.

Comment 9: [Limitations: You said food insecurity was not considered. Why? Some research used minimum dietary diversity as proxy for food insecurity. Why have you not used this?”].

Response: Thank you for this comment. We would like to clarify that minimum dietary diversity was included in our analysis as a child-level covariate. However, in the Limitations section we stated that food insecurity was not directly measured, as the DHS does not collect comprehensive household food insecurity indicators across all surveys included in this study.

While minimum dietary diversity can serve as a proxy for aspects of food access and feeding practices, it does not fully capture household-level or chronic food insecurity. We have revised the Limitations section to clarify this distinction, explicitly noting that dietary diversity was included in the analysis, but that the absence of a direct measure of food insecurity remains a limitation.

Reviewer 2:

Comment 1:

[ “Line 346 table 2,

Please carefully recheck and revise the percentage calculations in the tables, as several variables do not sum to 100%, including maternal literacy, maternal anemia, drinking water, sanitary facilities, birth order, and cluster altitude.

You have specified the anemia cutoff for children under five; however, the cutoff used to define maternal anemia has not been clearly stated”].

Response: Thank you for this detailed observation.

1. Percentage calculations: All percentages in Table 2 have been rechecked and corrected. Discrepancies were due to rounding, which have now been corrected.

2. Maternal anemia cutoff: We have now clearly specified in the Methods section that maternal anemia was defined according to WHO criteria as hemoglobin <12 g/dL for non-pregnant women and <11 g/dL for pregnant women, following standard DHS procedures.

Comment 2:

[ “Line 501-505

The observed inverse association between the lack of vitamin A supplementation and the co-occurrence of anemia and undernutrition should be clearly explained and contextualized in the Discussion section, with reference to existing evidence”].

Response: Thank you for this important concern.

We have revised the Discussion section to provide a clearer explanation and contextualization of the observed inverse association between lack of vitamin A supplementation and the co-occurrence of anemia and undernutrition.

We explain that this finding arises from the MGWR coefficient estimates, reflecting spatially varying relationships rather than a uniform effect across the study area. In certain locations, the inverse association likely reflects programmatic targeting, where vitamin A supplementation is more intensively provided in high-risk populations with a higher underlying burden of anemia and undernutrition. Importantly, this result should not be interpreted as an effect of withholding vitamin A supplementation, but rather as an indicator of complex, place-specific healthcare delivery patterns and residual confounding factors.

Comment 3:

[Line 512-516

The inverse association between recent diarrhea and the co-occurrence of anemia and undernutrition is contrary to existing evidence. This finding should be critically explained and discussed in the Discussion section].

Response: Thank you again for this important comment.

While recent diarrhea was positively associated with the co-occurrence of anemia and undernutrition in most areas, some regions showed a negative beta coefficient, indicating an inverse association. The inverse association is a true local relationship revealed because MGWR does not force a single global average. It shows that the diarrhea-malnutrition link is context-dependent. We argue that this localized inverse association does not invalidate the known pathophysiology but rather reveals how place-specific contextual factors can modify and even reverse a population-level relationship.

Most importantly, A key strength of the Multiscale Geographically Weighted Regression (MGWR) model is its ability to reveal that relationships between variables vary across space. This means a factor like recent diarrhea can be a strong risk factor in one community, have no association in another, and, as our analysis shows, even demonstrate an inverse association in specific locations. The finding of a significant negative local coefficient in certain clusters is a direct outcome of this analytical approach. It indicates that in these specific geographical areas, a higher local prevalence of recent childhood diarrhea was associated with a lower local prevalence of the co-occurrence of anemia and undernutrition. This

---

## [Editor Report · Decision Letter 1]

19 Jan 2026

Dear Dr. Kassaw,

Thank you for submitting your revised manuscript to PLOS ONE. After careful consideration, we find your manuscript to be scientifically suitable for PLOS ONE. However, before proceeding further we would invite you to submit a revised version of the manuscript addressing the point raised by the academic editor (see below).

We look forward to receiving your revised manuscript.

Kind regards,

Neetu Choudhary, PhD

Academic Editor

PLOS One

Journal Requirements:

**Additional Editor Comments:**

1.) As identified under comment 7 of reviewer 1 and explained by your response, you have included Zimbabwe under East Africa,based on the United Nations geoscheme for Africa, which groups countries into regional classifications for statistical and analytical purposes. Please add this justification for country classification in the section on study setting in the manuscript.

---

## [Author Response · Author response to Decision Letter 2]

20 Jan 2026

Additional Editor Comment:

Please add justification for including Zimbabwe under East Africa in the study setting section.

Response:

Thank you for this suggestion. We have now added a justification for the inclusion of Zimbabwe in the Study Setting section of the manuscript.

Location of revision:

Methods and materials section - Data source and study setting.

All requested files, including the revised manuscript with tracked changes, the clean manuscript, and this response letter, have been uploaded accordingly.

---

## [Editor Report · Decision Letter 2]

22 Jan 2026

Spatial Distribution and Factors Associated with Co-occurrence of Anemia and Undernutrition among Children Aged 6-59 Months in East Africa

PONE-D-25-34030R2

Dear Dr. Kassaw,

We’re pleased to inform you that your manuscript has been judged scientifically suitable for publication and will be formally accepted for publication once it meets all outstanding technical requirements.

Kind regards,

Neetu Choudhary, PhD

Academic Editor

PLOS One
---

## [Editor Report · Acceptance letter]

PONE-D-25-34030R2

PLOS One

Dear Dr. Kassaw,

I'm pleased to inform you that your manuscript has been deemed suitable for publication in PLOS One. Congratulations! Your manuscript is now being handed over to our production team.

Kind regards,

on behalf of

Dr. Neetu Choudhary

Academic Editor

PLOS One